# Evaporative water loss of 1.42 million global lakes

Gang Zhao[1,2], Yao Li [2], Liming Zhou [3] & Huilin Gao [2✉]

The evaporative loss from global lakes (natural and artificial) is a critical component of the terrestrial water and energy balance. However, the evaporation volume of these water bodies —from the spatial distribution to the long-term trend—is as of yet unknown. Here, using satellite observations and modeling tools, we quantified the evaporation volume from 1.42 million global lakes from 1985 to 2018. We find that the long-term average lake evaporation is $1500 \pm 150$ km$^3$ year$^{-1}$ and it has increased at a rate of 3.12 km$^3$ year$^{-1}$. The trend attributions include an increasing evaporation rate (58%), decreasing lake ice coverage (23%), and increasing lake surface area (19%). While only accounting for 5% of the global lake storage capacity, artificial lakes (i.e., reservoirs) contribute 16% to the evaporation volume. Our results underline the importance of using evaporation volume, rather than evaporation rate, as the primary index for assessing climatic impacts on lake systems.

[1] Department of Global Ecology, Carnegie Institution for Science, Stanford, CA 94305, USA. [2] Zachry Department of Civil and Environmental Engineering, Texas A&M University, College Station, TX 77843, USA. [3] Department of Atmospheric and Environmental Sciences, State University of New York at Albany, Albany, NY 12222, USA. ✉email: hgao@civil.tamu.edu

Covering about 5 million km$^2$ of the Earth's land area, lakes (natural and artificial) are key components of global ecological and hydrological systems[1–4]. Lakes support aquatic and terrestrial biodiversity, and are an important water resource for humans[5,6]. Due to their large open water areas—and the strong vapor pressure gradient at the water-atmosphere interface—lakes can lose a massive volume of water through evaporation (i.e., latent heat flux)[7,8]. The dynamics of lake evaporative water loss depend on water area and evaporation rate, both of which vary by geographical location and are sensitive to the manifestations of a complex changing environment[9]. For instance, evaporation rate can be altered by warming temperatures[10,11] and by elevated solar radiation[8], while open water areas can increase from shrinking lake ice cover[12] or decrease from extreme drought conditions[13]. Thus, it is crucial to understand the spatiotemporal changes and drivers of evaporative water loss from lakes for better aquatic ecosystem and water resources management.

However, due to a dearth of reliable, globally consistent, and locally practical datasets, evaporative water loss has not been quantified at a global scale. In the past, the accuracy of open water area and evaporation rate estimations has been hindered by various challenges such as cloud contamination of satellite images[14,15], lake heat storage quantification[16], and lake ice duration modeling[17,18]. The existing global studies have mostly focused on evaporation rate changes (solely)[11,16], and not on the overall evaporation volume. However, without factoring in the lake area dynamics and the lake ice freeze/thaw cycles, the evaporation rate alone cannot represent the magnitude of lake water loss. These approaches are thus inadequate for lake water/energy balance assessment and water resources management. Several local to regional studies have provided more reliable estimates[7,8,19], but extrapolating these results globally is inappropriate due to the large spatial heterogeneity of evaporation rate and water area. Consequently, the roles of global lakes within climate systems cannot be fully evaluated, as evaporation is the sole process linking energy exchanges and water cycles.

Here, we present the first global lake evaporation volume (GLEV) dataset, which contains the monthly evaporative water loss information of 1.42 million lakes (≥10$^5$ m$^2$) from 1985 to 2018. These lakes include both natural lakes and artificial lakes (referred to as "reservoirs" hereafter). For each lake and each month, the evaporation volume ($V_E$) was calculated as a function of the evaporation rate ($E_{lake}$), the lake surface area ($A_s$), and the fraction of ice duration ($f_{d,ice}$). In particular, the heat storage changes for lakes—which have been typically overlooked due to the complexities and difficulties in considering them—were quantified to improve the accuracy of the evaporation rate estimation[8]. The monthly lake surface areas were reconstructed using a Landsat-based global surface water dataset[14], and the monthly fractions of ice duration were modeled using air temperature and freeze/thaw lag information[18] (see Methods). The lake open water areas ($A_o$)—which exclude the lake ice cover—were calculated as the product of lake surface area ($A_s$) and the fraction of open water duration (i.e., 1-$f_{d,ice}$). To assess the long-term trend of each variable, the monthly results were aggregated to annual bases in the subsequent calculations.

## Results

**Spatial heterogeneity of lake evaporation volume.** In total, the annual $V_E$ of global lakes (excluding the Caspian Sea) from 1985 to 2018 is estimated to be 1500 ± 150 km$^3$ (Fig. 1), which is 15.4% higher than the previous model-based estimate (i.e., 1300 km$^3$)[20]. The spatial distribution of $V_E$ (Fig. 1a) is mostly linked to the $A_o$ distribution, but is also affected by $E_{lake}$ (Supplementary Fig. 1). For example, the five Laurentine Great Lakes and the seven

African Great Lakes contribute 8.8% and 15.7% to global $V_E$, while their corresponding total area is 9.1% and 6.2% (of the global total). In the region above 40°N, lakes contribute 62% of global $A_o$, but generate 46% of global $V_E$. The majority (83%) of the $V_E$ in this region occurs in the period from June to November (Fig. 1b)—i.e. the summer months in the Northern Hemisphere—which have the least lake ice coverage and the highest evaporation rates. The differences between the $A_s$ and $A_o$ distributions also confirm that the impact of lake ice on $A_s$ is more predominate over the high-latitude smaller lakes (Fig. 1c). The average annual $A_o$ for global lakes from 1985 to 2018 is $1.47 \times 10^6$ km$^2$, which is about 63% of $A_s$ ($2.34 \times 10^6$ km$^2$), indicating that a large portion of the global lake area is covered by ice. With regard to lake $V_E$, it first increases and then decreases at the same time as $E_{lake}$ increases (Fig. 1d). For lakes with $E_{lake}$ < 1500 mm year$^{-1}$, the $A_o$ and $E_{lake}$ jointly contribute to the exponentially increasing $V_E$. For lakes with $E_{lake}$ > 1500 mm year$^{-1}$, the $A_o$ tends to be limited by a drier climate, resulting in a reduction in $V_E$ despite the increase of $E_{lake}$ (Supplementary Fig. 1). This also suggests that using the evaporation rate alone cannot accurately represent the impacts of climate change on evaporative water loss due to the spatial mismatch between the distributions of $E_{lake}$ and $A_o$ (Fig. 1d and Supplementary Fig. 1). Instead, $V_E$ is the direct metric of such loss, and thus a better index of climate change for purposes of water resources management.

While the total lake surface area only accounts for 1.57% of the global land area, lake evaporation plays an important role in global and regional terrestrial evapotranspiration, as manifested through the $V_E/V_{ET}$ ratio. The land evapotranspiration volume was quantified by multiplying the Moderate Resolution Imaging Spectroradiometer (MODIS) Global Terrestrial Evapotranspiration product (i.e., MOD16A2) to the land surface area, and then the total evapotranspiration volume ($V_{ET}$) was derived by adding the lake evaporation volume to the land evapotranspiration volume. In total, lake evaporation contributes 2.37% to the global terrestrial $V_{ET}$. This $V_E/V_{ET}$ ratio exhibits a large range for different river basins (Fig. 2). Basins with large lake systems or in arid regions tend to have higher percentage values. For example, the Great Lakes Basin in North America has an $V_E/V_{ET}$ value of 27%, and the Tigris-Euphrates River Basin has a value of 13%. Conversely, due to the large soil evaporation and vegetation evapotranspiration, humid basins tend to have low percentage values (e.g., the Amazon River Basin has a $V_E/V_{ET}$ ratio of 0.5%).

**Long-term trends of lake evaporation volume.** For most of the nine thermal regions[21], the lake evaporation volume has increased during the past 34 years due to both an increasing evaporation rate and an increasing lake area (Fig. 3 and Supplementary Table 1). The $E_{lake}$ values for all of the nine regions show significant increasing trends (p-value < 0.05; Mann-Kendall non-parametric test). Specifically, $E_{lake}$ in the Northern Frigid (NF) region has risen the most—3.7% decade$^{-1}$. On average, the global $E_{lake}$ is increasing at a rate of 1.5% decade$^{-1}$ from 1985 to 2018, primarily due to increasing air temperature and vapor pressure deficit (Supplementary Fig. 2). For $A_o$, most regions have also shown an increasing trend. The interannual variability of $A_o$ generally follows the pattern of the regional climatological and hydrological variations[22]. For example, the rapid decrease of $A_o$ in the Southern Warm (SW) region from 2000 to 2009 (−18% decade$^{-1}$, −1030 km$^2$ year$^{-1}$) is attributed to the severe Australia Millennium drought[13]. Meanwhile, due to decreasing lake ice associated with warming, the NF region has shown a steady increase of $A_o$ (3.8% decade$^{-1}$, 760 km$^2$ year$^{-1}$). The long-term trend patterns of $V_E$ over different regions are generally consistent with their $A_o$ patterns, but are also modified by their $E_{lake}$ trends. The largest

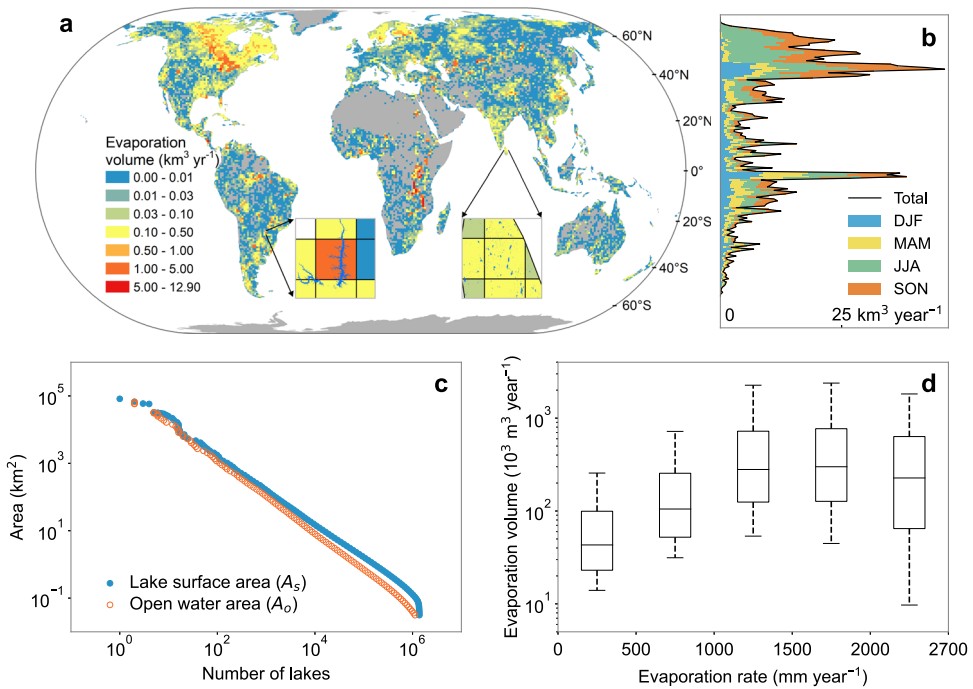

**Fig. 1 Spatial distributions of global lake evaporation volume. a** Gridded annual evaporation volume ($V_E$) averaged from 1985 to 2018. **b** Latitudinal distribution and seasonal variability of $V_E$. **c** Distribution of lake surface area ($A_s$) and open water area ($A_o$). **d** Statistical distribution of $V_E$ at different evaporation rate ($E_{lake}$) ranges. The original results for 1.42 million lakes and reservoirs were aggregated to equal-area grids in panel **a** under the World Eckert IV projection for purposes of better illustration. The areas of gray in the maps indicate no data. Seasons in panel **b** are represented by DJF (December, January, and February), MAM (March, April, and May), JJA (June, July, and August), and SON (September, October, and November). Box plots in panel **d** show upper quartile, median, and lower quartile, and the whiskers show the extreme value ranges.

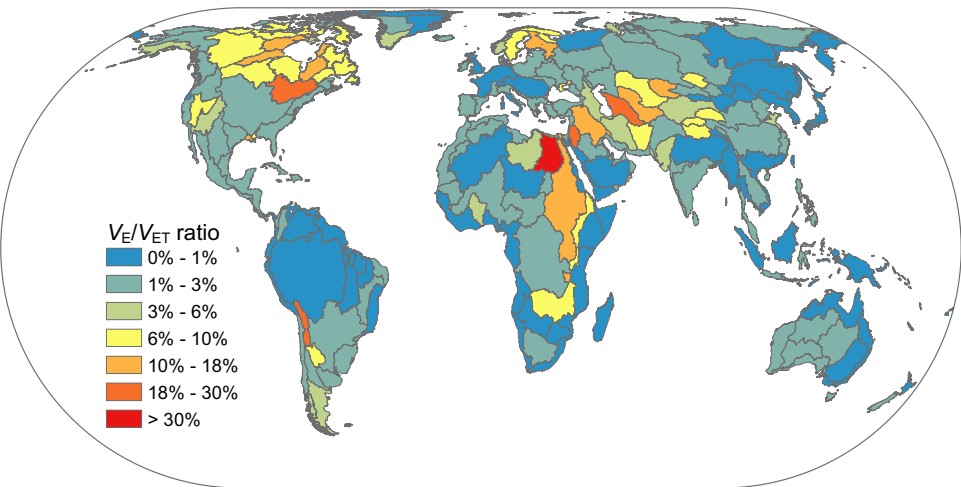

**Fig. 2 Percentage of lake evaporation volume ($V_E$) versus total evapotranspiration volume ($V_{ET}$) over the period from 2001 to 2018 for each HydroSHEDS level-3 basin.** Total $V_{ET}$ is defined as the sum of lake evaporation volume and land evapotranspiration volume, with the latter calculated by multiplying the evapotranspiration rate from the MOD16A2 product by the corresponding land area.

increasing trend (4.5% decade$^{-1}$, 0.6 km$^3$ year$^{-1}$) is again in the NF region, due to both the increasing $E_{lake}$ and $A_o$. Globally, lake evaporation volume ($V_E$) has been increasing at a rate of 2.1 ± 1.6% decade$^{-1}$ (i.e., 31.2 ± 24 km$^3$ decade$^{-1}$; see Supplementary Fig. 3 for uncertainty).

Compared to natural lakes, the evaporative water losses and their associated trends from reservoirs are more pronounced. According to the GLEV, the 6715 artificial reservoirs from HydroLAKES contribute 16% (235 km$^3$ year$^{-1}$) to the global evaporation volume—even though they only account for 5% of the storage capacity, and 10% of the surface area, of all lakes

(natural and artificial) combined. This quantity of reservoir evaporative loss is equivalent to 20% of the global annual consumptive water use (1185 km$^3$)[23]. From 1985 to 2018, evaporative water loss from reservoirs has been increasing at a rate of 5.4% decade$^{-1}$, which largely outpaces the global trend for all 1.42 million lakes (i.e., 2.1% decade$^{-1}$).

**Attributions of trends and variability of lake evaporation volume.** Three factors, $E_{lake}$, $f_{d,ice}$, and $A_s$, have contributed to the long-term trend of lake evaporation volume (see line plot in

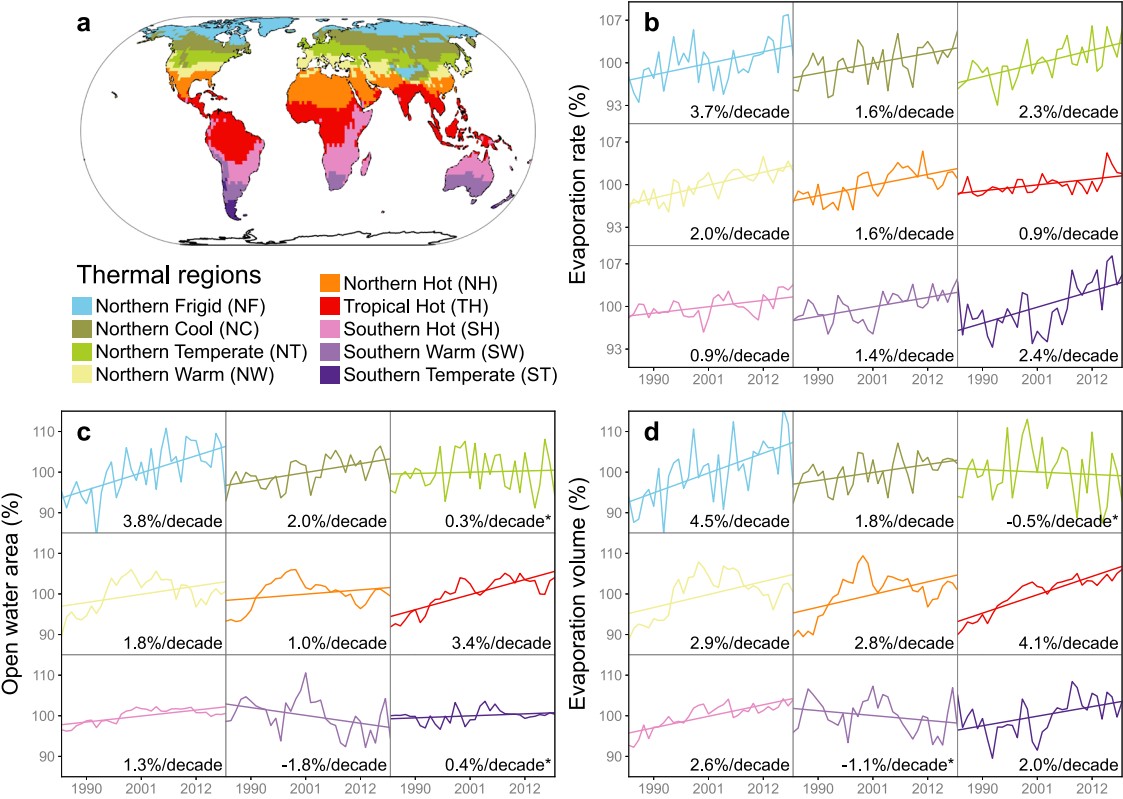

**Fig. 3 Long-term trends of global lake evaporation. a** The nine lake thermal regions defined in Maberly et al.[21] **b** Long-term trends of lake evaporation rate ($E_{lake}$) for the nine regions. **c** Long-term trends of lake open water area ($A_o$) for the nine regions. **d** Long-term trends of evaporation volume ($V_E$) for the nine regions. Each sub-panel shows the annual time series from 1985 to 2018—and its trend—for the corresponding thermal region. All values for $E_{lake}$, $A_o$, and $V_E$ are shown as a percentage (which is the actual value divided by the long-term averaged value). The trend indicated in each sub-panel was calculated based on the annual percentage values using a linear regression. Values without an asterisk are significant ($p$-value < 0.05) and values with an asterisk are not.

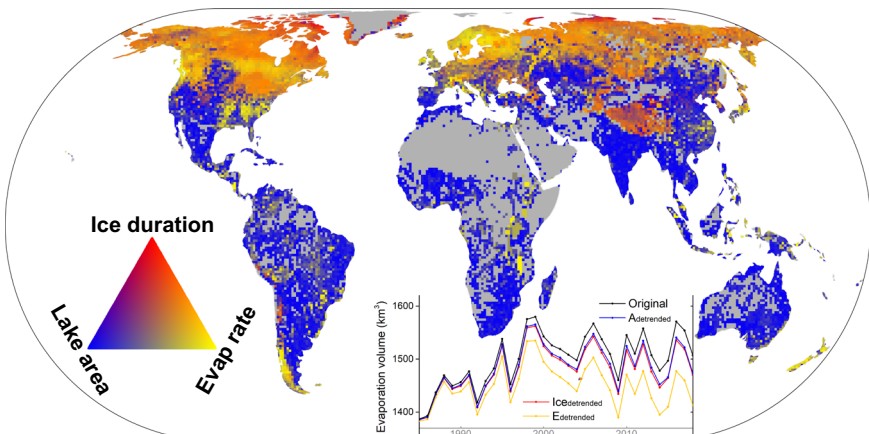

**Fig. 4 Attributions of long-term trends and interannual variability of lake evaporation volume.** The map shows the relative contributions from evaporation rate ($E_{lake}$), lake surface area ($A_s$), and lake ice duration ($f_{d,ice}$) to the interannual variability of evaporation volume ($V_E$). The inset line graph shows the original $V_E$ time series and the three $V_E$ time series plots calculated from the detrended $A_s$, $f_{d,ice}$, and $E_{lake}$, respectively. The gray areas in the map indicate no data. The "Original" line represents the $V_E$ time series from the global lake evaporation volume (GLEV) dataset. The "$A_{detrended}$" line represents the $V_E$ time series obtained by detrending the surface area time series for each lake. Similarly, the "$Ice_{detrended}$" and the "$E_{detrended}$" lines show the $V_E$ time series obtained after detrending the ice duration and the evaporation rate time series (respectively) for each lake.

Fig. 4). By detrending each of the three factors and then calculating the difference between the corresponding $V_E$ time series and the original $V_E$ time series (see Methods), we quantified that the contributions of $E_{lake}$, $f_{d,ice}$, and $A_s$ to the trend of $V_E$ are 48 km³ year⁻¹ (58%), 19 km³ year⁻¹ (23%), 16 km³ year⁻¹ (19%), respectively. Each of these three factors has contributed to

the increasing trend of $V_E$ positively, with the rising $E_{lake}$ being the dominant factor.

The percentage contribution of each of the three factors ($E_{lake}$, $f_{d,ice}$, and $A_s$) to the interannual changes of the lake evaporation volume has shown a clear spatial pattern (Fig. 4; see Methods). Based on the primary drivers of the interannual variability of $V_E$,

**Table 1 The three meteorological datasets used in this study.**

| Dataset | Spatial resolution | Temporal resolution | Developing institute | Link |
|---|---|---|---|---|
| TerraClimate[30] | 1/24° | Monthly | University of Idaho | http://www.climatologylab.org/terraclimate.html |
| ERA5[31] | 1/4° | Monthly | European Centre for Medium-Range Weather Forecasts | https://www.ecmwf.int/en/forecasts/datasets/reanalysis-datasets/era5 |
| GLDAS[32] | 1/4° | Monthly | NASA Goddard Space Flight Center | https://ldas.gsfc.nasa.gov/gldas |

we grouped the 1.42 million global lakes into two categories. For the first category, the $V_E$ variability of a lake is dominated by non-precipitation forcings. For instance, an increase of global air temperature can elevate $E_{lake}$[11] and reduce $f_{d,ice}$[12]. There are 1.09 million lakes (77%) that belong to this category, with a combined total open water area of $1.13 \times 10^6$ km². The climatic effect is especially notable in the high-latitude (e.g., Canada and northern Eurasia) and high-altitude (e.g., Tibetan Plateau) regions. Given the amplification effect of climate change in these regions[24], we expect an accelerated increase of $V_E$ in the future. It is also worth noting that for humid regions (e.g., the southeastern U.S. and southeastern China), the interannual variability of $V_E$ is mainly affected by $E_{lake}$ changes (because lake surface areas in these regions are mostly stable[15]). For the second category, the interannual variability of $V_E$ is mainly controlled by regional hydrologic conditions (e.g., surface runoff) and/or reservoir operations—both of which are the direct drivers for $A_s$ changes[22]. There are 0.33 million lakes that belong to this category, and they are scattered across a large portion of the global land area. For these lakes, the annual changes in $A_s$ prevail over the variabilities of $E_{lake}$ and/or $f_{d,ice}$. For instance, arid and semi-arid regions (e.g., the western U.S., southern Africa, and Australia) are typically affected by multiyear wet-dry cycles, and thus $V_E$ is highly correlated with $A_s$. For relatively humid regions (e.g., India), interannual variability from both precipitation and water use can substantially affect $A_s$ and then $V_E$. It is worth noting that 63% of all reservoirs (4250 out of the 6715 reservoirs in HydroLAKES) belong to this category. This underlines the importance of incorporating area dynamics for accurately estimating evaporative water loss to improve water management.

## Discussion

Our findings have significant environmental, societal, and economic implications, as the global evaporative loss will be accelerated in the future under global warming. As sentinels of climate change impact, lake systems respond quickly in terms of lake ice phenology and evaporation. Given the large increase of temperature in the high-latitude and high-altitude regions[24], the loss of lake ice can result in greater heat uptake (as the liquid water surface has a much lower albedo than ice) and larger open surface areas to evaporate. Together, the consequent greater evaporation flux to the atmosphere can alter both air humidity and local/regional hydrological processes, and thus needs to be more accurately described in Earth System Models. In addition, the substantial reservoir evaporation volume and its trend (5.4% decade⁻¹) under a warming climate can impose a hefty strain on water resources as demand for agricultural, industrial, and domestic water continues to increase in the context of population growth.

To the best of our knowledge, the GLEV is the very first dataset to provide long-term monthly time series evaportion values for 1.42 million individual natural lakes and reservoirs worldwide. With area values from Landsat high-resolution satellite observations and evaporation rates from a validated physically-based model for each lake, the dataset is globally consistent and locally practical. This freely available dataset can be beneficial to the wider science community and decision-makers. For example, this dataset can complement accurate water availability estimation, which especially needs to be prioritized during droughts[25]. Information about the transfer of water vapor from lakes into the atmosphere can improve simulations of moisture transport and recycling[26]. While several pioneering global studies have focused on surface water extent[14], lake and river ice[12,27], lake surface temperature[10], and surface water storage[4], our analysis contributes to the growing body of knowledge about global water bodies—and shows how lake systems are responding to the ongoing climate change.

## Methods

**Lake masks.** Shapefiles for the global lakes (natural and artificial) were obtained from the HydroLAKES dataset[28], which contains 1,427,688 water bodies that are 0.1 km² or larger. In this study, we included all of these water bodies except for the Caspian Sea. Among these lakes, HydroLAKES identified 6715 reservoirs based on the Global Reservoir and Dam database (GRanD)[29]. To include all possible historical water coverage scenarios, we buffered the HydroLAKES dataset based on lake area (Supplementary Table 2).

**Meteorological data.** To account for the uncertainties associated with the meteorological data, three representative datasets were used (Table 1): TerraClimate[30], ERA5[31], and the Global Land Data Assimilation System (GLDAS)[32]. These three widely used datasets were developed independently by different institutes/agencies based on a wide range of different input sources, and thus can represent the meteorological forcing conditions as well as their uncertainties. Monthly data values of the four governing variables for our evaporation rate algorithm—surface downward shortwave radiation, surface air temperature, surface humidity, and wind speed—were collected from these three datasets. Specifically, because surface downward shortwave (solar) radiation is one of the most important energy terms for the lake energy balance[8,33], we have independently validated its values using ground-based measurements from the Global Energy Balance Archive (GEBA)[34]. The $R^2$-values (i.e., 0.95, 0.94, 0.93 for Terra-Climate, ERA5, and GLDAS) and the biases (i.e., −0.57, 7.11, −2.63 W m⁻²) (Supplementary Fig. 4) have indicated that the reanalysis incident solar radiation data used are of good quality.

To calculate the time series of the governing variables, the average values from the grids that intersect individual lakes were calculated for each lake and each month. Thus, for each dataset, the monthly time series values of these four variables were generated from Jan 1985 to Dec 2018 for 1,427,687 lakes. Because GLDAS does not have a single product that covers the entire period, we concatenated the reprocessed GLDAS-2.0 data from 1985 to 2014 and the GLDAS-2.1 data from 2015 to 2018 (with GLDAS-2.1 bias-corrected to GLDAS-2.0).

**Evaporation volume.** The monthly time series of evaporation volume from Jan 1985 to Dec 2018 was calculated for each lake. For a given lake, its monthly evaporation volume was estimated using Eq. 1.

$$V_E = E_{lake} \times A_s \times \left(1 - f_{d,ice}\right)/1000 \qquad (1)$$

where $V_E$ is the evaporation volume (m³ d⁻¹); $E_{lake}$ is the evaporation rate for each month (mm d⁻¹); $A_s$ is the monthly surface area (km²), and $f_{d,ice}$ is the monthly fraction of ice duration (which is defined as the time percentage of a month when a lake is fully covered by ice). In particular, the monthly $E_{lake}$ time series was calculated based on a newly developed algorithm[8] using monthly meteorological data, the monthly $A_s$ time series was estimated using a Landsat-based global surface water dataset (GSWD)[14], and the monthly $f_{d,ice}$ time series was modeled using reanalysis air temperature and freeze/thaw lags. The calculation of each of these variables is detailed in the following sections.

**Evaporation rate**. The monthly evaporation rate time series for the 1.42 million lakes were calculated based on a newly developed algorithm[8], which is based on the Penman combination equation (Eq. 2) but explicitly quantifies the heat storage changes (Eq. 3).

$$E_{lake} = \frac{\triangle(R_n - \delta U) + \gamma\lambda(a + bu_2)L_f^{-0.1}(e_s - e_a)}{\lambda(\triangle + \gamma)} \quad (2)$$

$$\delta U = \rho_w c_w \bar{h} \frac{T_w - T_{w0}}{\triangle t} \quad (3)$$

where $E_{lake}$ is the lake evaporation rate (mm d$^{-1}$); $\Delta$ is the slope of the saturation vapor pressure curve (kPa °C$^{-1}$); $R_n$ is the net radiation (MJ m$^{-2}$ d$^{-1}$); $\delta U$ is the heat storage change of the water body (MJ m$^{-2}$ d$^{-1}$); $a$ and $b$ are the wind function coefficients (and are equal to 2.33 and 1.65, respectively)[35]; $u_2$ is the screen height (2 m) wind speed (m s$^{-1}$); $L_f$ is the average fetch length of the water body (m); $e_s$ is the saturated vapor pressure at air temperature (kPa); $e_a$ is the air vapor pressure (kPa); $\lambda$ is the latent heat of vaporization (MJ kg$^{-1}$); $\gamma$ is the psychrometric constant (kPa °C$^{-1}$); $\rho_w$ is the density of water (kg m$^{-3}$); $c_w$ is the specific heat of water (MJ kg$^{-1}$ °C$^{-1}$); $\bar{h}$ is the average water depth (m); $T_w$ is the water column temperature at the current time step (°C); $T_{w0}$ is the water column temperature at the previous time step (°C); and $\Delta t$ is the time step (set as 30 days in this study). By incorporating the "generally applicable" wind function from McJannet et al.[35], our algorithm eliminates any parameter calibration process and only needs four governing meteorological variables (surface downward shortwave radiation [MJ m$^{-2}$ d$^{-1}$], air temperature [°C], vapor pressure deficit [kPa], and wind speed [m s$^{-1}$]), the monthly lake fetch ($L_f$), and the average lake depth ($\bar{h}$) for the $E_{lake}$ calculation.

For each lake, the forcing data were averaged over the lake surface area from the three meteorological forcing datasets (TerraClimate, ERA5, and GLDAS). The monthly lake fetch values[8] were calculated using (1) National Centers for Environmental Prediction (NCEP) wind direction climatology data[36], (2) the shapefiles from the HydroLAKES dataset[28], and (3) the lake area dynamics (as explained in the following section). For each lake, the actual epilimnion thickness—which determines the $\delta U$—was derived using the smaller value between the calculated potential epilimnion thickness based on surface area[37] and the average lake depth from the HydroLAKES dataset[28].

This newly developed algorithm has been intensively validated in the U.S. region in our previous work[8]. Additional validations were implemented on a global scale by adding four international lakes that have reliable evaporation observations (Supplementary Fig. 5). These results show that the incorporation of heat storage simulation can significantly improve the accuracy—especially for deep lakes (which have larger heat storage capacities than shallow lakes). Furthermore, we have systematically evaluated the energy balance and aerodynamic terms using observations at Lake Taihu[38,39] (Supplementary Fig. 6) and Lake Mead[40] (Supplementary Fig. 7). Our evaluations for both shallow and deep lakes indicate that our algorithm is robust, and that the good agreement of evaporation rate simulation is not caused by the cancellation of errors. Also, our results are consistent with those of other modeling/observational studies focused on several extremely large lakes (Supplementary Table 3). Generally, the evaporation estimation for extremely large lakes (>10,000 km$^2$) is hindered by (1) scaling limitations associated with observational methods (e.g., eddy covariance), (2) ignoring heat storage changes, and (3) the inaccuracy of meteorological data from lake surfaces. Thus, we only compared the long-term average evaporation values for these large lakes. In addition, comparisons of lake evaporation rate with actual and potential evapotranspiration over land (Supplementary Fig. 8) show a high degree of spatial consistency. The patterns of the evaporation rate zonal mean (i.e., latitudinal distribution) are also similar to the results from Wang et al.[11].

**Lake surface area dynamics**. The monthly water area time series for the 1.42 million lakes from 1985 to 2018 was reconstructed based on a combination of the dynamic Landsat-based global surface water dataset (GSWD)[14] and static HydroLAKES shapefiles[28]. The GSWD is a global "water body" product[41] that was derived from Landsat imagery via an expert system that considers numerous factors (e.g., cloud, shadow, terrain shadow, lava). However, remotely sensed "water body" does not represent the "lake" water extent. Thus, we further used Hydro-LAKES shapefiles as the outer boundaries of lakes to ensure that extracted water pixels from GSWD are from lakes. This type of approach has been commonly adopted for lake area time series estimations[42–44]. GSWD contains multiple products with different temporal resolutions: monthly, annual, monthly climatology, and static. One caveat of the GSWD monthly product is that its global water maps contain large areas of "no data" pixels[14], which result from multiple factors, including limited Landsat coverage (especially before 1999), cloud, cloud shadow, terrain shadow, and the Scan Line Corrector failure of Landsat-7. Therefore, direct extraction of the area time series from these monthly maps can lead to severe underestimations.

Previously, we developed a robust image enhancement algorithm to reduce the impacts of the "no data" pixels on water area extraction when using the GSWD monthly product[15]. However, this algorithm requires extensive computational resources and is not practical for generating area time series for 1.42 million lakes. Therefore, in this study, we updated the algorithm to (1) consistently reduce the impacts of image contamination (e.g., cloud and shadow) on the extracted monthly

water area; (2) improve the computational efficiency so as to be practical at the million-lake processing level; and (3) better characterize the seasonality in data-limited years. The updated algorithm was developed using the GSWD annual product with seasonality reconstructed using long-term monthly mean area values. Based on the original monthly product, GSWD also provides the annual composite images, which classify each pixel for each given year into one of four categories: (1) year-round water, (2) seasonal water, (3) not water, and (4) no data. These annual water classification maps have significantly eliminated the number of "no data" pixels—but meanwhile, also conceal the monthly dynamics. Building upon these annual maps, we reconstructed the monthly area dynamics for each year (Fig. 5) based on the long-term monthly mean area values from the original monthly global maps (Eqs. 4 and 5).

$$A_R(y, m) = A_{y,ya} + W_m \times A_{y,ss} \quad (4)$$

$$W_m = \frac{\overline{A_m} - \min_{i=1\ldots12} \overline{A_i}}{\max_{i=1\ldots12} \overline{A_i} - \min_{i=1\ldots12} \overline{A_i}} \quad (5)$$

where $A_R(y, m)$ is the reconstructed area for year $y$ and month $m$ (km$^2$); $A_{y, ya}$ is the year-around area for year $y$; $A_{y, ss}$ is the seasonal area for year $y$; and $W_m$ (ranging from 0 to 1) is the weighting factor for month $m$ (ranging from 1 to 12). $\overline{A_m}$ is the average area for month $m$ (which is calculated based on the original monthly global maps from 1985 to 2018; km$^2$); $\min_{i=1\ldots12} \overline{A_i}$ is the minimum monthly average area (km$^2$); and $\max_{i=1\ldots12} \overline{A_i}$ is the maximum monthly average area (km$^2$). Specifically, the monthly average is calculated by stacking all of the historical clear water pixels together to get the water occurrence image for the given month, and then using it as a weighted image to calculate the total area. When a specific year had no $A_{y, ya}$ and/or no $A_{y, ss}$ due to limited Landsat coverage during the early years (especially before 1999, Supplementary Fig. 9), these values were linearly interpolated using valid values from adjacent years. Although these linearly interpolated values may not represent the true water areas with perfect accuracy, they were derived from the best available data from Landsat. Furthermore, we can expect that the missing values will have a limited impact on the 34-year trend at a large scale. This is because the missing coverage of Landsat in the early years is mainly located in the high-latitude regions (Supplementary Fig. 9), where the area changes are primarily affected by ice coverage (simulated by air temperature that is described in the following section).

The validation of this simple but computationally practical (for 1.42 million lakes) method was implemented by comparison to area values extracted from the original cloud-free monthly maps from the Global Reservoir Surface Area Dataset (GRSAD)[15]. The overall coefficient of determination ($R^2$) for the 6715 reservoirs involved is 0.97 (Fig. 6). This method performs better for relatively larger lakes. For example, the $R^2$-value for the lakes between 1 km$^2$ to 100 km$^2$ is 0.99, while it is 0.17 for the lakes <0.01 km$^2$. Given that the majority of the lakes in the HydroLAKES dataset (96%, Supplementary Table 2) have area values >0.1 km$^2$, the proposed method is thus believed to be capable of providing reliable reconstructed monthly area values. In addition, we calculated the error statistics for each of the 6715 reservoirs (Fig. 6b, c). The median relative bias (rBias) of 1.6% and median relative Root Mean Square Error (rRMSE) of 9.2% further indicate the high quality of the reconstructed monthly area values.

The reconstructed monthly area values were further compared with in-situ observed storage/elevation time series values (Supplementary Fig. 10) and satellite altimetry data wherever available (Supplementary Fig. 11). Compared to GRSAD, the new results perform better during years with fewer Landsat observations (e.g., 1992 to 1998 for Mossoul Lake, Iraq). Overall, the reconstructed areas yield comparable data quality as those of the GRSAD monthly area values, and are capable of representing the area dynamics for lakes with a wide range of sizes.

**Lake ice duration**. The ice coverage of a lake depends on its climatic, geographic, morphological, and bathymetric characteristics[17,18]. In particular, air temperature plays a dominant role in lake ice phenology[45]. However, a simple 0 °C isothermal approach cannot accurately represent the freeze and thaw events of lakes. This is because the energy stored in lake water before winter delays the formation of ice, and the ice thickness build-up during the winter slows down the rate of ice melting[18]. Specifically, the freeze lag (i.e., time lag between a 0 °C frost day and lake freeze-up) is primarily affected by the internal heat of the lake, which is correlated to the lake's depth. With regard to thaw lag (i.e., the time lag between a 0 °C warm day and lake ice break-up day), it is mainly determined by the ice thickness which depends on the average winter temperature.

Based on the long-term lake ice observation records from the Global Lake and River Ice Phenology Database [GLRIPD][46], we established (1) the relationship between the freeze lag and the average lake depth; and (2) the relationship between the thaw lag and the average winter (December, January, and February) temperature (Fig. 7). The air temperature data were calculated from the TerraClimate monthly dataset, and then were linearly interpolated to a daily time step to estimate the 0 °C isotherm dates. Due to the large variability in actual daily air temperature data, reconstructed daily temperatures based on monthly time series can provide more reliable 0 °C isotherm dates[47]. The average lake depth data were collected from the HydroLAKES dataset[28]. To derive more representative

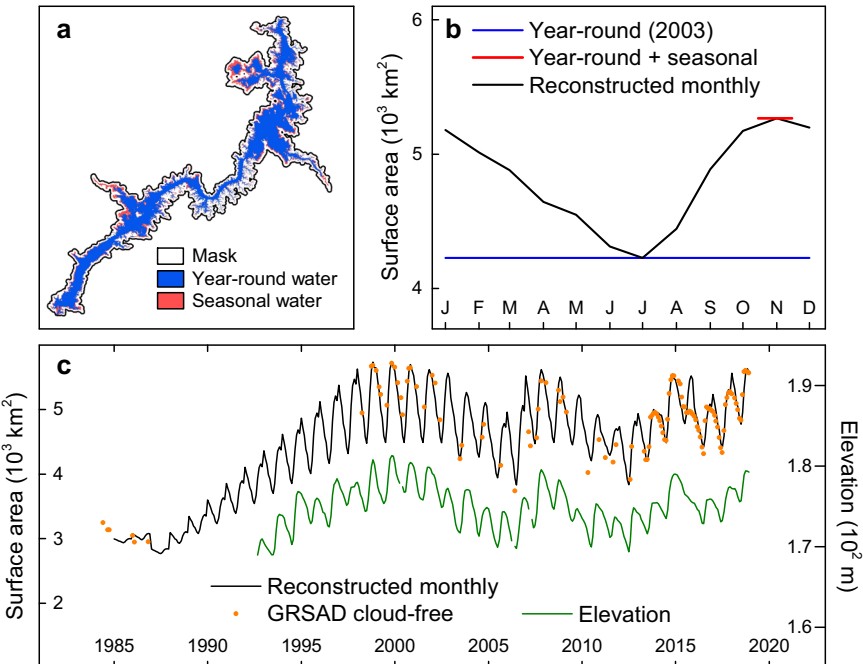

**Fig. 5 Reconstruction of the monthly lake surface area time series based on GSWD annual water classification maps.** Lake Nasser in Egypt in the year 2003 is used as an example to show the reconstruction approach. **a** The GSWD annual water classification map for 2003. **b** The reconstructed monthly area for 2003. **c** The monthly time series of the reconstructed area from Jan 1985 to Dec 2018 obtained after applying the reconstruction approach for each year. The cloud-free water surface area data were collected from the Global Reservoir Surface Area Dataset (GRSAD) and the elevation data were collected from the Global Reservoirs and Lakes Monitor (G-REALM) dataset. Both time series were used as references for comparison with the reconstructed monthly area values.

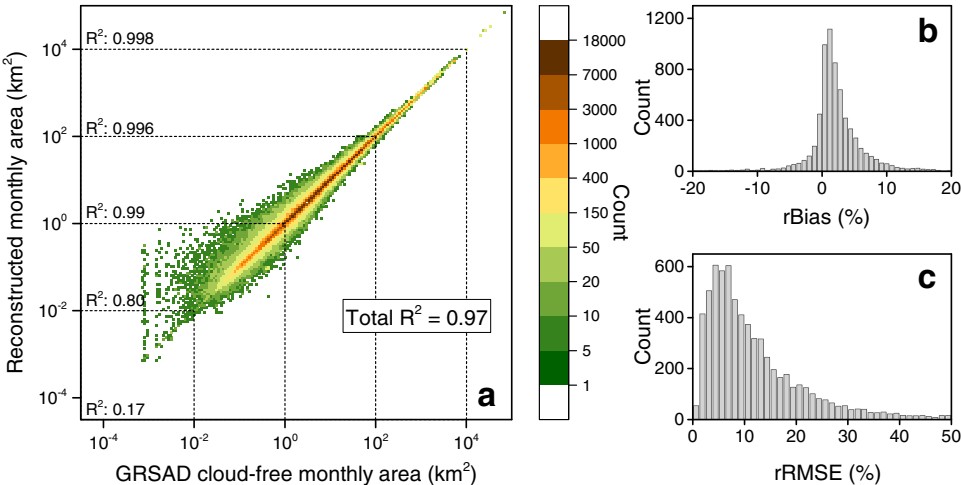

**Fig. 6 Validation of reconstructed monthly surface area values by comparing to GRSAD for 6715 reservoirs. a** Comparison between reconstructed monthly areas and cloud-free monthly areas from GRSAD. **b** The histogram of relative bias (rBias) for each of the reservoirs. **c** The histogram of relative root-mean-square error (rRMSE) for each of the reservoirs.

relationships for estimating freeze lag and thaw lag, we (1) calculated the freeze/thaw lag days for each observed freeze/thaw date at each lake; (2) averaged the freeze and thaw lag days for each lake; and (3) fitted the equations using these data for all of the lakes. An exponential relationship was found to best fit the freeze lag data, and a linear relationship was found to best fit the thaw lag data (Fig. 7).

First, these two fitted equations (Fig. 7b, c) were evaluated using simulated annual ice duration for lakes that have observations for both ice-on and ice-off dates (Fig. 7d). For the 76 such lakes, the $R^2$-value is 0.93 and the standard deviation of the absolute bias is 9.2 days year$^{-1}$. Considering the low $E_{lake}$ during the freezing and thawing periods, we expect that less uncertainty will be propagated to the $V_E$. Second, the monthly simulation performance of these two fitted equations (Fig. 7b, c) was evaluated using long-term ($\geq$ 20 years) in-situ ice phenology data for 14 North American lakes (Supplementary Fig. 12). The average

bias for these lakes is 7 days year$^{-1}$ (with a range from $-19$ to 33 days year$^{-1}$), indicating satisfactory data quality. It is worth noting that our quantification of ice duration simplified the lake ice phenology by assuming binary values (i.e., ice or no ice) for each day. For large lakes, knowing the true areal fraction of ice coverage can be important. However, by aggregating the daily binary values to monthly floating percentage values, the impacts of such simplification can be reduced. For example, validation for the North American Great Lakes (Supplementary Fig. 13) shows a high level of agreement between simulated annual ice fraction (averaged $f_{d,ice}$) and observed annual ice fraction (derived from daily measurements of ice coverage).

We did not use remotely sensed ice cover (e.g., MODIS) to represent the ice phenology for two reasons: (1) The tradeoff between spatial and temporal resolution significantly limits the wide application of satellite ice observations, especially for small lakes; (2) Cloud cover—and reflectance similarity between

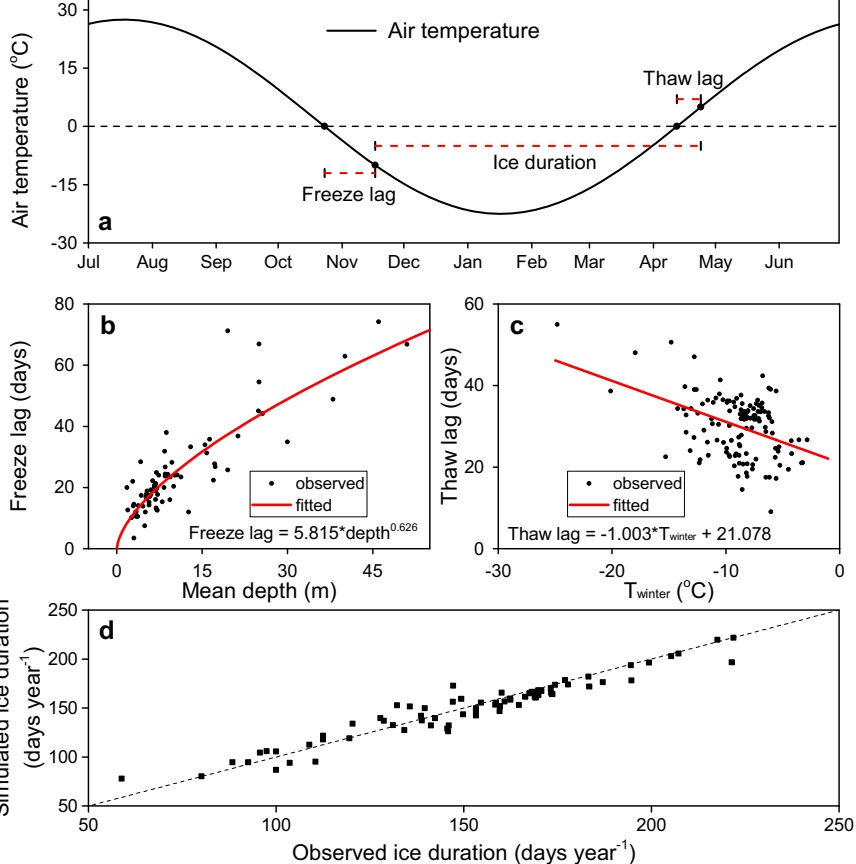

**Fig. 7 Simulation of freeze lag and thaw lag days. a** A conceptual scheme to simulate the ice duration. **b** The fitted relationship between freeze lag days and lake average depth. **c** The fitted relationship between thaw lag days and average winter temperature. **d** Comparison between simulated and observed annual ice duration, with a root-mean-square error (RMSE) of 9.2 days year$^{-1}$.

cloud and surface ice/snow—reduces the data coherence of the ice phenology time series (Supplementary Fig. 14). To ensure the data homogeneity, reanalysis air temperature is commonly used to smooth the ice cover time series from satellite data. However, such a method then relies on the data quality of reanalysis air temperature, leading to a similar performance with our modeling approach[48]. Nonetheless, satellite sensors (e.g., Landsat and MODIS) are still highly promising for constructing continuous lake ice phenology, especially by fusing multiple sensors and machine learning algorithms.

**Trend and variability attribution**. Trend attribution of the increasing lake evaporation volume was implemented by detrending each of these three factors (i.e., $E_{lake}$, $f_{d,ice}$, and $A_s$)[49,50]. For instance, to calculate the global $V_E$ time series with the area trend removed, we (1) detrended the $A_s$ time series for each lake while keeping the $E_{lake}$ and $f_{d,ice}$ time series unchanged, (2) multiplied these three terms together to get the new $V_E$ time series for each lake, and (3) aggregated the results to obtain the global $V_E$ time series. The contributions of each factor ($E_{lake}$, $f_{d,ice}$, and $A_s$) were then calculated as the long-term difference between the original $V_E$ time series and the detrended ones. The relative contributions were subsequently calculated as the percentage values of the individual actual contributions over the sum of the actual contributions.

For each lake, the interannual variability of $V_E$ is affected by the variability of three components: $E_{lake}$, $f_{d,ice}$, and $A_s$. To evaluate the relative importance of these three contributing factors, we adopted the following approach: First, we calculated the $R^2$-values between the annual $V_E$ and each of these three factors (i.e., $E_{lake}$, $f_{d,ice}$, and $A_s$). Then, we aggregated the $R^2$ results (using the average lake areas as weighting factors) within each equal-area grid for $V_E$-$E_{lake}$, $V_E$-$f_{d,ice}$, and $V_E$-$A_s$. Next, the three $R^2$-values were normalized using a sum-of-unity to show the global distribution of the relative importance of these three factors (Fig. 4). We then grouped the lakes into two categories based on the major drivers of the $V_E$ variability: (1) non-precipitation forcings; and (2) terrestrial hydrologic conditions and reservoir operations. A lake was assigned to the first category when the $R^2$-value for $V_E$-$E_{lake}$ or $V_E$-$f_{d,ice}$ was larger than that for $V_E$-$A_s$. Conversely, when the $R^2$-values for $V_E$-$E_{lake}$ and $V_E$-$f_{d,ice}$ were both smaller than that for $V_E$-$A_s$, the lake was assigned to the second category.

**Sources of uncertainty and algorithm caveats**. Uncertainty in the $V_E$ estimate is propagated from that in the $E_{lake}$, $f_{d,ice}$, and $A_s$. For $E_{lake}$, the input forcing generally carries large levels of uncertainty[51]. This uncertainty was quantified using three independent reanalysis datasets—TerraClimate, ERA5, and GLDAS—and is described as the standard deviation of the relative bias (SDRB) for the evaporation rate ($E_{lake}$) of each lake (by regarding average values as the truth values). This calculation leads to an uncertainty value of 7.22%. With respect to ice duration ($f_{d,ice}$), its uncertainty is mainly inherited from the air temperature uncertainty (Fig. 7a) and the regression uncertainty (Fig. 7d). The air temperature uncertainty of the reanalysis datasets originates from multiple sources, including the data assimilation algorithm and various sources related to assimilated data (i.e., satellites and ground stations). Here, using the same method as described above, we found that the air temperature uncertainty (from the three reanalysis datasets) can lead to an uncertainty of 1.45% for $f_{d,ice}$. For the regression component, the associated uncertainty can be calculated using the SDRB presented in Fig. 7d, resulting in a value of 2.52%. Similarly, the surface area ($A_s$) has an uncertainty of 6.1% based on the information presented in Fig. 6. In total, by assuming a normal distribution for these four types of uncertainty and running a million cycles of a Monte Carlo simulation, the total uncertainty was estimated to be 9.93%, yielding an uncertainty range for global $V_E$ of about 150 km$^3$ year$^{-1}$.

There are a few caveats that are worth noting. A full energy balance under ice conditions is not explicitly simulated when calculating $E_{lake}$. The interaction between radiation, ice, and water makes lake heat storage complicated, which then extends to the evaporation process[19]. Due to the small $E_{lake}$ during the winter months, we expect a limited impact on the overall $V_E$ and its long-term trend. However, this warrants the need for collecting more $E_{lake}$ observational data during months when the lakes are partially covered by ice for model evaluation. In addition, sublimation from lake ice can also contribute to lake water loss during the winter season (Supplementary Note 1). However, due to the insulation effect of the overlying snow cover[52] and the relatively small sublimation rate from ice/snow[53,54], we would expect only a limited effect for this process on the overall lake water loss (Supplementary Fig. 15). With regard to lake surface area, the limited Landsat global coverage during its early years affects the data quality of the GSWD. This, in turn, affects our area estimations. However, due to the usage of the reconstruction algorithm (Eqs. 3 and 4), such impacts—especially on long-term trends—can be reduced.

## Data availability

The shapefiles of HydroLAKES were downloaded from https://www.hydrosheds.org/page/hydrolakes. The data links to the three meteorological datasets (i.e., TerraClimate, ERA5, and GLDAS) can be found in Table 1. The surface water area time series were processed on the Google Earth Engine using the JRC annual water maps (https://developers.google.com/earth-engine/datasets/catalog/JRC_GSW1_1_YearlyHistory). The GRSAD data were downloaded from https://doi.org/10.18738/T8/DF80WG. The in situ observed lake ice coverage data were downloaded from https://nsidc.org/data/lake_river_ice/. The processed global lake evaporation volume (GLEV) dataset—containing monthly lake open areas, evaporation rates, and evaporative water loss data for 1.42 million lakes—is available at https://doi.org/10.5281/zenodo.4646621.

## Code availability

The codes for the core algorithm and the main figures are available at https://github.com/gzhaowater/lakeEvap.

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

## Acknowledgements

This research was supported by the National Aeronautics and Space Administration (NASA) grants 80NSSC18K0939 and 80NSSC17K0358. It was also partially supported by the US Department of Energy (DOE) Water Power Technologies Office as a part of the SECURE Water Act Section 9505 Assessment. L.Z. was supported by National Science Foundation (NSF) AGS-1952745 and AGS-1854486. This research has benefitted from the usage of the Google Earth Engine platform.

## Author contributions

G.Z. conceptualized the study, developed the method, performed analysis, and drafted the manuscript. H.G. conceptualized the study, assisted with the analysis, and reviewed and edited the manuscript. Y.L. and L.Z. assisted with the analysis and reviewed and edited the manuscript.

## Competing interests

The authors declare no competing interests.
