## [Peer Review File · Nature Communications]

Evaporative water loss of 1.42 million global lakesReviewers' Comments:

Reviewer #1:

Remarks to the Author:

The authors of this study provided an estimate of evaporative water loss from global lakes. In addition to determining the total evaporation volume, they performed an attribution analysis, by partitioning the detected trend in evaporation volume to changes in lake area and changes in evaporation rate. Changes in lake area were further divided into a climate-related component (e.g., reduced ice period) and an anthropogenic component (e.g., reservoir expansion). A key conclusion is that lakes contributed a disproportionate amount – disproportion relative to their areal fraction – to global land evapotranspiration.

A strength of this study is that it used a high-resolution (both temporal and spatial) lake extent dataset to drive evaporation calculation. From this the authors were able to map spatial patterns of lake evaporative loss and quantify how these patterns evolved over time. I also like the fact that they separated the evaporative volume of reservoirs from that of natural water bodies.

I have detected several weaknesses. My main misgiving concerns their model of evaporation rate. This is an empirical model requiring heavy calibration. (1) Is it appropriate to apply the wind function of McJannet to global lakes? That function was calibrated against a small amount of data from small lakes. Its functional form is counter-intuitive: it implies weaker wind as lake size increases, but we know from observational evidence that wind should be stronger over a larger lake; (2) The empirical relationships for longwave radiation developed by Mohseni and Stefan are meant for streams and rivers, whose turbulent mixing is very different from that occurring in lake water; (3) Their procedure for determining T_e contains an important error: the heat storage term is left out of their equation for the latent heat flux (Eq 6, their RME paper). Since T_e was used to estimate heat storage and lake heat storage is a large component of the lake energy balance at sub-annual time scales, I am left wondering about the overall reliability of their result.

Their calculations were restricted to open-water periods, and evaporative loss from ice-covered lakes was omitted. I don't think this appropriate because ice evaporation is not negligible for high-latitude lakes. I suspect that this omission could cause a number of problems: the evaporation volume could be biased low, the role of area change could be exaggerated, and the rising trend of evaporation rate could be too strong.

They used reanalysis meteorological fields to drive their model calculations. A problem common to all reanalysis data products is that the incident shortwave radiation is biased high. This problem was not discussed.

According to the authors, their modeled evaporation rate seems to be in agreement with values observed for a handful of lakes. It is not clear if this agreement was a result of calibration, or if different errors got canceled out. A true demonstration of model robustness requires validation against a set of data that is independent from the set used for model calibration.

In view of my concerns about, I am not comfortable recommending this paper for publication. If the authors want to continue this line of research, perhaps they should switch to a more mechanistic model of evaporation, such as the modified Priestley-Taylor model (Yang & Roderick, Q. J. R. Meteorol. Soc. 145, 1118–1129). Another alternative is to run a physically-based model, such as that by Subin et al. (J. Adv. Model. Earth Syst. 4, M02001), in offline mode driven by reanalysis meteorology and interfaced with the authors' lake extent (and depth) data.

Minor comments:

L69: RE ice duration: have you tried satellite-based ice phenology?

L149: How did you differentiate man-made reservoirs and natural water bodies? Satellites can isolate water pixels from land pixels but they cannot differentiate water types.

L162: "non-precipitating forcings": I am not sure if this is an accepted technical jargon

Eq 2: units are not consistent

Supplementary Figure 1: It would be interesting to compare your rate estimate (spatial pattern, zonal mean) with those published by other researchers.

Reviewer #2:

Remarks to the Author:

Review of Evaporative water loss of 1.42 million global lakes by Zhao et al.

The idea of this study is to estimate evaporation volume of global lakes from 1985 to 2018, which is ambitious. First, a global lake data from 1985 to 2018 is necessary. At present, 1.42 million global lakes from HydroLakes are produced based on STRM DEM in February, 2000, there is no available a long-term global lake dataset. The authors used global waterbody products, which is greatly different from lakes, especially impossible to acquire at monthly scale. The title of 1.42 million global lakes could be misled. In addition, monthly ice duration modeled using air temperature had large uncertainties. The ratio of lake evaporation water loss to global terrestrial evapotranspiration is not comparable as different data and spatial resolution. In addition, the writing of this manuscript is not readable, which is not followed by NC style. Overall, the data used can not support for estimates of monthly global lake evaporation changes from 1985 to 2018.

Major comments:

1. In this study, the authors estimated the monthly evaporative water loss for 1.42 million lakes from 1985 to 2018. Some concepts such as "monthly" and "lakes" are not right. The global waterbody products are used directly, although it is produced at monthly scale, the gaps are large as no available data because of cloud contamination of Landsat images. In addition, the global water products are different from lakes. The authors claimed a "1.42 million lakes", I understand a global lake mask produced in 2000 using SRTM DEM water body product (<https://doi.org/10.1038/ncomms13603>) was used. However, lakes are different from waterbody products, it is not possible to produce a global lake product, especially at monthly scale. This is the core of this study as the authors examined the trends of evaporative water loss, and the novel for global lakes.
2. The monthly fraction of ice duration was modeled using air temperature and freeze/thaw lags. Again, I want to know the reliability of monthly fraction of ice duration. For a global scale, air temperature in station sparse regions has large uncertainties. This could result in modelled fraction of ice duration with large uncertainties. Especially, the simulation was conducted at monthly scale.
3. The writing of this manuscript is not readable. It is okay for Abstract and Introduction from Line 40-62. However, after Line 63, too many numbers and formulas. The structure of this manuscript did not flow the style of Nature Communications, and need rewrite. Especially, the results and discussion should be separated from methods clearly.
4. The analysis of increasing evaporation linking with lake ice coverage and lake area is simple. It is not a quantitative evaluation, and only calculated by turning off one of variables.

Specific comments:

- "reservoirs" is manmade lakes? How to differentiate lakes from reservoirs?
- "The monthly lake surface area was reconstructed using a Landsat-based global surface water dataset17" The authors did not understand the global waterbody product well, it cannot be used as a global lake product or constructed product, especially at a monthly scale.

- "This value is 15.4% higher than previous estimates..." How you concluded that your value is higher than previous study. It is compared in same study period and condition?
- "To quantify the VE/ET ratio, we calculated the land ET using the MODIS...". MODIS data has a coarse resolution relative to 30-m Landsat water body products, and moreover, MODIS data is available after 2000, but Landsat water product used is available from 1985. However, you calculated the ratio of VE/ET ratio, it is not reasonable.
- How to understand 1.57 mm/yr/yr?
- "Globally, the long-term trend for VE is 2.1%/decade (3.12±??? km³/yr)." The uncertainties of trend should be included. Other values are same problems.
- Reservoirs from HydroLakes is a static dataset. How you calculated reservoir evaporative loss from 1985 to 2018?
- Validation of reconstructed monthly surface area by compared to 6715 reservoirs. The high R² does not mean high quality, which is determined by number of samples. More importantly, the mapped reservoir areas for comparison are from same date?
- The simulation of lake ice phenology is simple. Actually, lake ice phenology can be observed from MODIS data. However, MODIS has a coarse resolution, which is incomparable to Landsat waterbody data used in this study.

Reviewer #3:

Remarks to the Author:

This manuscript is tremendous effort that provides the first volumetric estimate of evaporation for global lakes while also generating an interesting global dataset. I believe the focus on water volume as opposed to rate is a valuable and original contribution. The methods are well described and results well communicated.

I have few suggestions to improve this manuscript. My suggestions mostly involve clarifying some writing in the abstract/introduction and 1 other paragraph. Also, there are a lot of results and information in this manuscript, therefore I encourage the authors to reduce some of the results if possible (but only if there is something they believe is less important for the paper). Detailed comments are below.

Abstract

Generally, I recommend using more clear language to articulate why this is important. Simply saying we don't know the distributions (and of what?) and that it is intrinsically complex is a bit vague. I think the volume aspect is a better "hook" for this paper, at least in the abstract. Stating simply, we don't know the volume of water lost from lakes because we focus on rates. And explain in 1 sentence why evaporative volume matters.

Line 22: delete "human".

Line 22 and 47: what is meant by distributions (of evaporative water loss)? Spatial variability or distribution, the distribution/histogram of all values of lake evaporation? Please clarify this word or remove to make more simple.

Line 23: What is intrinsic complexity? Please use simple language for the abstract or consider not using this word intrinsic complexity.

Main

Line 42: artificial lakes support ecosystems as well. Perhaps keep it to a more general word like... "Lakes support aquatic and terrestrial biodiversity and are an important water resource for humans"

Line 46: Consider taking a sentence or two to explain how the climate change impacts in parentheses impact evaporative loss rather than just listing them. Particularly, consider explaining the Bowen ratio in plain language.

Line 48-49: I agree it is crucial, but I am not sure how understanding evaporation mitigates climate

and support sustainable ecosystems. This is a great paper and analysis, and I do not think it requires vague justifications. Consider just saying (after you just explain a bit about how climate change is impacting evaporative loss through the 3 mechanisms in parentheses), "It is crucial to understand trends and drivers of evaporative water loss from lakes at a global scale."

Paragraph 2 is great. I think the argument for evaporative volume taking into account the surface water and ice dynamics is the strength of this work .

Line 144: This is very interesting! That reservoirs contribute more water loss.

Line 160: Consider saying "In addition to" rather than "besides".

Line 172: This doesn't seem like topic sentence for what this paragraph is about.

Line 173: 77% of lakes have exactly this area? Are less than this area? Or combined have this area? Please clarify this sentence.

172-178: This paragraph needs some attention. Each sentence seems to be about a different result and topic. Please clarify this paragraph or remove results/sentences in it that are less important to the story.

Line 206: "physically-based" model not physical-based.

Line 214: remove "sheds some critical light on" and change to "...and shows how lake systems are responding..."

John Gardner

REVIEWER COMMENTS

Reviewer #1 (Remarks to the Author):

The authors of this study provided an estimate of evaporative water loss from global lakes. In addition to determining the total evaporation volume, they performed an attribution analysis, by partitioning the detected trend in evaporation volume to changes in lake area and changes in evaporation rate. Changes in lake area were further divided into a climate-related component (e.g., reduced ice period) and an anthropogenic component (e.g, reservoir expansion). A key conclusion is that lakes contributed a disproportionate amount – disproportion relative to their areal fraction – to global land evapotranspiration.

A strength of this study is that it used a high-resolution (both temporal and spatial) lake extent dataset to drive evaporation calculation. From this the authors were able to map spatial patterns of lake evaporative loss and quantify how these patterns evolved over time. I also like the fact that they separated the evaporative volume of reservoirs from that of natural water bodies.

R1C0: We greatly appreciate the time and effort that the reviewer has put in reviewing our manuscript. We also thank the reviewer for identifying the strengths of our work and for suggesting improvements to our manuscript. In the following sections, we have provided point-to-point responses regarding your comments and made corresponding changes in the main manuscript. We hope that the improved manuscript can help the readers to better understand our study.

I have detected several weaknesses. My main misgiving concerns their model of evaporation rate. This is an empirical model requiring heavy calibration. (1) Is it appropriate to apply the wind function of McJannet to global lakes? That function was calibrated against a small amount of data from small lakes. Its functional form is counter-intuitive: it implies weaker wind as lake size increases, but we know from observational evidence that wind should be stronger over a larger lake;

R1C1: Thank you for the comment. We would like to clarify that our evaporation rate algorithm does not rely on any parameter calibration. The main equation for our evaporation algorithm (Zhao and Gao, 2019) is based on the Penman combination equation (Eq. R1C1-1)

$$E = \frac{\Delta(R_n - G) + \gamma f(u)(e_s - e_a)}{\lambda_v(\Delta + \gamma)} \quad (\text{R1C1-1})$$

It combines the energy budget and aerodynamic mass transfer to calculate the evaporation rate of open water. Compared to energy balance or mass transfer equations, the Penman equation eliminates the need for water skin temperature and provides robust evaporation estimation (Dingman, 2015). By adopting the wind function (Eq. R1C1-2) from McJannet et al. (2012), there are no calibrated parameters associated with our calculation. Here we would also like to clarify that we did not calibrate the wind function parameters based on the in-situ data we collected (Supplementary Fig. 5). Instead, we used the “*generally applicable*” parameters directly

from McJannet et al. (2012). In McJannet et al. (2012), the authors compiled lakes ranging from 0.07 m² to 33.5 km² (these area values were later converted to lake fetch) from a variety of geographic and climatic regions to calculate the wind function parameters.

$$f(u_2) = \lambda_v(2.33 + 1.65u_2)L_f^{-0.1} \quad (\text{R1C1-2})$$

Such function implies that wind function value (indicating the capability of wind to transfer water vapor from water surface to air) decreases with the increase of lake fetch (or area). This is because air humidity increases when moving downwind direction from the shore due to the entrainment of water vapor. There is no implication that wind speed decreases for a larger size of lakes. In fact, we used the wind speed directly from the reanalysis datasets (TerraClimate, ERA5, and GLDAS). Because the wind function from McJannet et al. (2012) was developed based on “*land-based meteorological data*”, land-based reanalysis data can be used for lake evaporation quantification.

We, therefore, applied the wind function from McJannet et al. (2012) for the global lakes and we believe it is appropriate because of two additional reasons. First, for the HydroLAKES datasets, there are 1427687 lakes in total and the majority (1422664, 99.6%) of these lakes are smaller than an area of 33.5 km² (maximum value from McJannet et al. 2012). Second, for larger lakes, we have also validated our algorithms using the observed evaporation rate (Supplementary Fig. 5 and Supplementary Table 3). Such agreement for extremely large lakes (> 1000 km²) can be explained by the exponential form of the wind function, which has a decreasing slope for large values of lake fetch (Fig. 2 in McJannet et al. 2012). For clarity, we have added the following clarifications in the new manuscript: “By incorporating the “generally applicable” wind function from McJannet et al. (2012), our algorithm eliminates any parameter calibration process and only needs four governing meteorological variables (surface downward shortwave radiation [MJ·m⁻²·d⁻¹], air temperature [°C], vapor pressure deficit [kPa], and wind speed [m·s⁻¹]), the monthly lake fetch (L_f), and the average lake depth (\bar{h}) for E_{lake} calculation.” (Lines 261-265).

Reference

- Dingman, S. L. (2015). *Physical hydrology*. Waveland Press.
- Zhao, G., & Gao, H. (2019). Estimating reservoir evaporation losses for the United States: Fusing remote sensing and modeling approaches. *Remote Sensing of Environment*, 226, 109-124.
- McJannet, D. L., Webster, I. T., & Cook, F. J. (2012). An area-dependent wind function for estimating open water evaporation using land-based meteorological data. *Environmental modelling & software*, 31, 76-83.

(2) The empirical relationships for longwave radiation developed by Mohseni and Stefan are meant for streams and rivers, whose turbulent mixing is very different from that occurring in lake water;

R1C2: The longwave radiation relationship was developed in Mohseni and Stefan (1999) by simplifying the Stefan–Boltzmann Law (in an exponential form; Eq. R1C2-1) to a linear equation (Eq. R1C2-2) when temperature value ranges from –30 to 50°C. However, such simplification is not affected by the turbulence state or the type (streams/rivers or lakes) of a water body (Dingman, 1972; Gianniou and Antonopoulos, 2007).

$$L^\uparrow = \varepsilon_w \sigma (T_e + 273.15)^4 \quad (\text{R1C2-1})$$

$$L^\uparrow \approx \varepsilon_w (kT_e + b) \quad (\text{R1C2-2})$$

where L^\uparrow is the outgoing longwave radiation of a water body; σ is the Stefan–Boltzmann constant; ε_w is the emissivity of water; T_e is the equilibrium temperature of the water body; k and b are empirical parameters developed by Mohseni and Stefan (1999).

Please also note that such empirical equation (Eq. R1C2-2) was only used for the equilibrium temperature calculation rather than being used for calculating energy balance in the Penman equation. When a water body is in the ideal equilibrium state, there are no changes of flux variables and thus the water column temperature is a constant value, which is defined as the equilibrium temperature (T_e). Then, the outgoing longwave radiation (L^\uparrow) can be simplified to Eq. R1C2-2 (Mohseni and Stefan, 1999) by assuming that water surface temperature equals water equilibrium temperature (Schmid et al., 2014).

Reference

- Dingman, S. L. (1972). Equilibrium temperatures of water surfaces as related to air temperature and solar radiation. *Water Resources Research*, 8(1), 42-49.
- Gianniou, S. K., & Antonopoulos, V. Z. (2007). Evaporation and energy budget in Lake Vegoritis, Greece. *Journal of Hydrology*, 345(3-4), 212-223.
- Mohseni, O., & Stefan, H. G. (1999). Stream temperature/air temperature relationship: a physical interpretation. *Journal of hydrology*, 218(3-4), 128-141.
- Schmid, M., Hunziker, S., & Wüest, A. (2014). Lake surface temperatures in a changing climate: a global sensitivity analysis. *Climatic change*, 124(1), 301-315.

(3) Their procedure for determining T_e contains an important error: the heat storage term is left out of their equation for the latent heat flux (Eq 6, their RME paper). Since T_e was used to estimate heat storage and lake heat storage is a large component of the lake energy balance at sub-annual time scales, I am left wondering about the overall reliability of their result.

R1C3: While the equilibrium temperature (T_e) is used for estimating heat storage, its derivation is independent of heat storage. This is because T_e is defined as the water temperature at which the sum of all heat fluxes through the water surface is zero, indicating that the water body and the above air reached an equilibrium/balanced state. In such an ideal condition under given radiative forcings, the water temperature is constant over time and there are no heat storage changes ($\Delta U = \rho_w c_w \bar{h} \frac{T_w - T_{w0}}{\Delta t} = 0$). Thus, under this ideal equilibrium condition, the Penman equation can be simplified to **Eq 6** (in the RSE paper) by not showing the zero ΔU .

Our calculation for T_e is consistent with its original definition (Edinger et al., 1968) and definitions used by other studies (Finch et al., 2001; Caissie et al., 2005; McMahon et al., 2013; Schmid et al., 2014). It is worth emphasizing that **Eq 6** is never used for calculating the actual lake evaporation rate. Rather, it is combined with Eqs 4,5,7 and Eq 8, only for the purpose of estimating water column temperature (T_w ; Eq 9 in the RSE paper) and heat storage change (ΔU ;

Eq 11 in the RSE paper). The final equation for calculating evaporation rate is still **Eq 1** (in the RSE paper), which explicitly contains the heat storage changes (ΔU) calculated by **Eq 11**.

In the new manuscript, we have added the formulation of heat storage as the new **Eq. 2** to show that heat storage is explicitly quantified. Also, we have clarified that “**The monthly evaporation rate time series for the 1.42 million lakes was calculated based on a newly developed algorithm (Zhao and Gao, 2019), which is based on the Penman combination equation (Eq. 2) but explicitly quantifies the heat storage changes (Eq. 3).**” (Lines 250-252).

Reference

- Caissie, D., Satish, M. G., & El- Jabi, N. (2005). Predicting river water temperatures using the equilibrium temperature concept with application on Miramichi River catchments (New Brunswick, Canada). *Hydrological Processes: An International Journal*, 19(11), 2137-2159.
- Edinger, J. E., Duttweiler, D. W., & Geyer, J. C. (1968). The response of water temperatures to meteorological conditions. *Water Resources Research*, 4(5), 1137-1143.
- Finch, J. W. (2001). A comparison between measured and modelled open water evaporation from a reservoir in south- east England. *Hydrological Processes*, 15(14), 2771-2778.
- McMahon, T. A., Peel, M. C., Lowe, L., Srikanthan, R., & McVicar, T. R. (2013). Estimating actual, potential, reference crop and pan evaporation using standard meteorological data: a pragmatic synthesis. *Hydrology and Earth System Sciences*, 17(4), 1331-1363.
- Schmid, M., Hunziker, S., & Wüest, A. (2014). Lake surface temperatures in a changing climate: a global sensitivity analysis. *Climatic change*, 124(1), 301-315.
- Zhao, G., & Gao, H. (2019). Estimating reservoir evaporation losses for the United States: Fusing remote sensing and modeling approaches. *Remote Sensing of Environment*, 226, 109-124.

Their calculations were restricted to open-water periods, and evaporative loss from ice-covered lakes was omitted. I don't think this appropriate because ice evaporation is not negligible for high-latitude lakes. I suspect that this omission could cause a number of problems: the evaporation volume could be biased low, the role of area change could be exaggerated, and the rising trend of evaporation rate could be too strong.

R1C4: In our study, the major focus is the evaporation process, which by definition is “*a type of vaporization that occurs on the surface of a **liquid** as it changes into the **gas** phase*”. Thus, we did not specifically quantify the sublimation (**solid to gas**) losses. While we agree with the reviewer that lake ice sublimation can also contribute to the water losses for high-latitude and high-altitude lakes, we believe the sublimation volume is generally much smaller than water evaporation from the lakes based on the following reasons.

- 1) In high-latitude and high-altitude regions, lake ice is mostly covered by snow in the winter season (Fig. R1C4). Such a snow layer can insulate the underneath ice and limit the direct heat transfer in the air-ice interface (Sturm and Liston, 2003). The accumulation of snow pressure in the snow-ice interface may also accommodate the formation of snow-ice and superimposed ice (Cheng et al., 2003; Semmler et al., 2012). These processes limit the lake ice sublimation rate. Meanwhile, ice sublimation can also be compensated by water vapor condensation when the atmospheric vapor pressure is larger than saturation pressure at ice temperature (Box and Steffen, 2001).

Fig. R1C4. a) Average snow depth from 1985 to 2018 for months that have an average below-zero air temperature from the ERA5 dataset; b) Kernel density of snow depths on the lake surface for the 1.42 million lakes.

- 2) Direct measurements of lake ice sublimation are complicated by (1) snow cover and snow sublimation, (2) atmospheric vapor condensation, and (3) ice ablation (Sturm and Liston, 2003; Box and Steffen, 2001). Thus, it is difficult to get reliable sublimation rate measurements for lake ice. Based on glacier snowpack measurements, several studies report an average snow sublimation rate of 0.47 mm/d (Reba et al., 2012; Sexstone et al., 2016; Stössel et al., 2010; Stigter et al., 2018; Wagnon et al., 2003). Because lake ice has a much smaller specific surface area (SSA) than snow crystals (Matzl and Schneebeli, 2006), we would expect that lake ice sublimation rate is much smaller, leading to little impact on lake water losses.

Because quantifying ice sublimation is beyond the scope of our water evaporation work, we have added such discussion in the manuscript to notify readers about such phenomenon that can also contribute to lake water losses: “In addition, sublimation from lake ice can also contribute to the lake water storage losses during the winter season. However, due to the insulation effect of overlaying snow cover (Sturm and Liston, 2003) and the relatively small sublimation rate from ice/snow (Sexstone et al., 2016; Reba et al., 2012), we would expect only a limited effect of such process on the overall lake water losses (Supplementary Fig. 13).” (Lines 433-436).

Reference

- Cheng, B., Vihma, T., & Launiainen, J. (2003). Modelling of superimposed ice formation and subsurface melting in the Baltic Sea. *Geophysica*, 39(1-2), 31-50.
- Box, J. E., & Steffen, K. (2001). Sublimation on the Greenland ice sheet from automated weather station observations. *Journal of Geophysical Research: Atmospheres*, 106(D24), 33965-33981.
- Dugan, H. A., Obryk, M. K., & Doran, P. T. (2013). Lake ice ablation rates from permanently ice-covered Antarctic lakes. *Journal of Glaciology*, 59(215), 491-498.
- Sturm, M., & Liston, G. E. (2003). The snow cover on lakes of the Arctic Coastal Plain of Alaska, USA. *Journal of Glaciology*, 49(166), 370-380.
- Semmler, T., Cheng, B., Yang, Y., & Rontu, L. (2012). Snow and ice on Bear Lake (Alaska)—sensitivity experiments with two lake ice models. *Tellus A: Dynamic Meteorology and Oceanography*, 64(1), 17339.

- Stössel, F., Guala, M., Fierz, C., Manes, C., & Lehning, M. (2010). Micrometeorological and morphological observations of surface hoar dynamics on a mountain snow cover. *Water Resources Research*, 46(4).
- Sexstone, G. A., Clow, D. W., Stannard, D. I., & Fassnacht, S. R. (2016). Comparison of methods for quantifying surface sublimation over seasonally snow-covered terrain. *Hydrological Processes*, 30(19), 3373-3389.
- Reba, M. L., Pomeroy, J., Marks, D., & Link, T. E. (2012). Estimating surface sublimation losses from snowpacks in a mountain catchment using eddy covariance and turbulent transfer calculations. *Hydrological Processes*, 26(24), 3699-3711.
- Stigter, E. E., Litt, M., Steiner, J. F., Bonekamp, P. N., Shea, J. M., Bierkens, M. F., & Immerzeel, W. W. (2018). The importance of snow sublimation on a Himalayan glacier. *Frontiers in Earth Science*, 6, 108.
- Matzl, M., & Schneebeli, M. (2006). Measuring specific surface area of snow by near-infrared photography. *Journal of Glaciology*, 52(179), 558-564.
- Wagnon, P., Sicart, J. E., Berthier, E., & Chazarin, J. P. (2003). Wintertime high-altitude surface energy balance of a Bolivian glacier, Illimani, 6340 m above sea level. *Journal of Geophysical Research: Atmospheres*, 108(D6).

They used reanalysis meteorological fields to drive their model calculations. A problem common to all reanalysis data products is that the incident shortwave radiation is biased high. This problem was not discussed.

R1C5: In our previous work (Zhao and Gao, 2019), we did notice the discrepancy of shortwave radiation among different reanalysis datasets. Such differences are mainly caused by the uncertainty in the cloud pattern modeling and aerosol/water vapor data (Urraca et al., 2018). For ERA5, several studies have reported comprehensive evaluations (He et al., 2021; Amjad et al., 2021) and recognized an overestimation of solar radiation despite that there is a significant improvement of ERA5 over ERA-Interim (an earlier version). For GLDAS and TerraClimate, because there are fewer studies that provide such evaluation, we chose to evaluate all of these three datasets together by ourselves.

Using the station-based solar radiation data collected from the Global Energy Balance Archive (GEBA; Wild et al., 2017), we evaluated the solar (shortwave) radiation from the three reanalysis datasets from all the ground stations (657 in total) that have data during our study period (Fig. R1C5). Overall, the reanalysis values agree very well with the observations, with R^2 ranging from 0.93 to 0.95. Keep in mind that the reanalysis data are area-averaged values for individual grid boxes ($1/24^\circ$ to $1/4^\circ$) while GEBA data are point measurements from individual stations. This scaling difference will result in some differences between modeled and observed, even for a perfect model, considering spatial heterogeneity in cloud and surface conditions. The absolute bias values confirm that ERA5 overestimated the solar radiation while TerraClimate and GLDAS slightly underestimated it. Specifically, the absolute bias of TerraClimate is less than 1 W/m^2 , showing its good performance. The mean bias value for these stations from the 3 reanalysis datasets is 4.88 W/m^2 , approximately equivalent to 0.107 mm/d , which is about 3.6% of the global average evaporation rate.

Based on our global evaluation, we believe the three reanalysis datasets overall agree well with the ground-based observations. Specifically, by using three reanalysis data, the uncertainty of such overestimation/underestimation can be quantified. We have added such information in the new manuscript: "Specifically, because surface downward shortwave (solar) radiation is one of the most important energy terms for lake energy balance (Zhao and Gao, 2019), we have independently validated its values using ground-based measurements from the Global Energy

Balance Archive (GEBA) (Wild et al, 2017). The R^2 values (i.e., 0.95, 0.94, 0.93 for TerraClimate, ERA5, and GLDAS) and the biases (i.e., -0.57, 7.11, -2.63 W/m^2) (Supplementary Fig. 4) indicate that the reanalysis incident solar radiation data used are of good quality.” (Lines 224-229).

Fig. R1C5. Validation of surface downward shortwave radiation from three reanalysis datasets, a) TerraClimate, b) ERA5, and c) GLDAS using d) 657 ground-based stations from GEBA.

Reference

- Urraca, R., Huld, T., Gracia-Amillo, A., Martinez-de-Pison, F. J., Kaspar, F., & Sanz-Garcia, A. (2018). Evaluation of global horizontal irradiance estimates from ERA5 and COSMO-REA6 reanalyses using ground and satellite-based data. *Solar Energy*, 164, 339-354.
- He, Y., Wang, K., & Feng, F. (2021). Improvement of ERA5 over ERA-Interim in simulating surface incident solar radiation throughout China. *Journal of Climate*, 34(10), 3853-3867.
- Amjad, M., Asim, M., Azhar, M., Farooq, M., Ali, M. J., Ahmad, S. U., ... & Hussain, A. (2021). Improving the accuracy of solar radiation estimation from reanalysis datasets using surface measurements. *Sustainable Energy Technologies and Assessments*, 47, 101485.

Wild, M., Ohmura, A., Schär, C., Müller, G., Folini, D., Schwarz, M., ... & Sanchez-Lorenzo, A. (2017). The Global Energy Balance Archive (GEBA) version 2017: A database for worldwide measured surface energy fluxes. *Earth System Science Data*, 9(2), 601-613.

According to the authors, their modeled evaporation rate seems to be in agreement with values observed for a handful of lakes. It is not clear if this agreement was a result of calibration, or if different errors got canceled out. A true demonstration of model robustness requires validation against a set of data that is independent from the set used for model calibration.

R1C6: Thank you. We would like to clarify that our evaporation rate algorithm does not rely on any parameter calibration. As detailed in R1C1, our algorithm is based on the Penman combination equation that needs 4 meteorological variables (shortwave radiation, air temperature, vapor pressure deficit, and wind speed) to calculate lake evaporation rate. Lake characteristics (i.e., lake depth and fetch length) were adopted/calculated from the HydroLAKES dataset. The in-situ data based on EC/BREB methods in Supplementary Fig. 5 were solely used to validate our algorithm instead of being used for any calibration. Because these in-situ data cover lakes from different geographic and climatic zones, our validation results clearly demonstrate the robustness of our algorithm. To our best knowledge, our dataset is validated over the most locations using **monthly** EC/BREB in situ data as compared to existing literature.

In view of my concerns about, I am not comfortable recommending this paper for publication. If the authors want to continue this line of research, perhaps they should switch to a more mechanistic model of evaporation, such as the modified Priestley-Taylor model (Yang & Roderick, Q. J. R. Meteorol. Soc. 145, 1118–1129). Another alternative is to run a physically-based model, such as that by Subin et al. (J. Adv. Model. Earth Syst. 4, M02001), in offline mode driven by reanalysis meteorology and interfaced with the authors' lake extent (and depth) data.

R1C7: We hope our responses above have clarified that our evaporation rate algorithm is a robust method to quantify open water evaporation. Here we would also like to address the differences between the Penman combination equation (which our algorithm is based on) and 1) the Priestley-Taylor equation (original and modified) and 2) aerodynamic mass transfer equations that most 1-D lake models (e.g., CLM4-LISSS, FLake, VIC-Lake) are based on.

The Penman combination equation combines the energy balance and aerodynamic mass transfer to eliminate the need for water surface temperature measurement (for full derivation see Dingman, 2015). The primary input meteorological forcings are radiation, air temperature, air humidity, and wind speed. Later after the Penman, Priestley and Taylor (P-T) simplified the Penman equation to remove the dependence on wind speed and introduced the empirical P-T coefficient (i.e., 1.26; Priestley and Taylor, 1972). However, such empirical coefficient is only appropriate for non-advective conditions and can change under advective conditions (Assouline et al., 2016; Eichinger et al., 1996). While Yang and Roderick (2019) improved the theoretical explanation of the P-T equation, they kept assuming that the heat storage term is zero (Section 2.2 from Yang and Roderick, 2019). Since heat storage is a crucial term in lake energy balance,

we believe results based on our method can better capture the physical representation of lake evaporation than the P-T and modified P-T.

A common formulation of latent heat flux in lake models is the atmospheric mass transfer equation that is based on Monin–Obukhov similarity theory (Table R1C7). However, the accuracy of such estimation is dependent on the accuracy of calculation of water surface temperature (for e_s^*) and water surface resistance/roughness (r_{aw} or z_{0q}). Here, we are not arguing that our method is unequivocally superior to other methods. In fact, all these methods contain uncertainties to some extent due to their respective assumptions and simplifications. For our algorithm, because we explicitly incorporated the heat storage simulation and have validated our algorithm over lakes from diverse geographic and climatic regions with reliable EC or BREB measurements, we are confident about its global applicability and robustness of the results.

Table R1C7. Formulation of different evaporation rate calculation methods.

Name	Main equation	Reference
This study	$\lambda E = \frac{\Delta(R_n - G) + \gamma \lambda f(u)(e_s - e_a)}{\Delta + \gamma}$	Zhao and Gao, 2019; Zhao et al., 2020
Priestley-Taylor	$\lambda E = 1.26 \frac{\Delta}{\Delta + \gamma} (R_n - G)$	Priestley and Taylor, 1972
Modified Priestley-Taylor	$\lambda E = \frac{\Delta}{\Delta + 0.24\gamma} (R_n - G)$	Yang and Roderick, 2019
CLM4-LISSS	$\lambda E = \frac{\lambda \rho_{air}}{r_{aw}} (e_s^* - e_a)$	Subin et al., 2012
FLake	$\lambda E = \frac{\lambda u_* v}{S_c \cdot (\ln(h/z_{0q}) + \psi)} (e_s^* - e_a)$	Mironov et al., 2010
VIC-Lake	$\lambda E = \frac{\lambda u \rho_{air} v^2}{\ln^2(h/z_{0q})} (e_s^* - e_a)$	Bowling and Lettenmaier, 2010

where λ is the latent heat of vaporization; E is the open water evaporation rate; Δ is the slope of the saturation vapor pressure curve; R_n is the net radiation; G is the heat storage change of the water body; γ is the psychrometric constant; $f(u)$ is the wind function; e_s is the saturated vapor pressure at air temperature; e_s^* is the saturated vapor pressure at water surface temperature; e_a is the air vapor pressure; ρ_{air} is air density; r_{aw} is the atmospheric resistance with respect to latent heat; u_* is the friction velocity; v is the von Karman constant; S_c is the Schmidt number at neutral stability; h is the height of measured air temperature and humidity; z_{0q} is the surface roughness length for latent heat; ψ is the Monin–Obukhov stability function for specific humidity profile. **Note** that this table is mainly used to demonstrate the formulation of different methods and the equations were directly adopted from the referenced publication, thus the units may not be consistent.

In addition, we recently developed our own lake model (LTEM; Zhao et al., 2020) based on remotely sensed (i.e., MODIS) water surface temperature and 1-D Hostetler model (Hostetler and Bartlein, 1990). This model has been used for generating the monthly evaporation

component of NASA's new MODIS/VIIRS global water reservoir product (Li et al., 2021; <https://modis-land.gsfc.nasa.gov/modgwr.html>). Although LTEM can simulate the detailed temperature profile dynamics, we only noticed a marginal improvement for evaporation rate estimation compared to the algorithm we used in this study (see Table 3 in Zhao et al., 2020). However, the new model requires time series of remotely sensed water surface temperature, for which data that is based on MODIS are too spatially coarse (i.e., 1km resolution). For Landsat, there are multiple limitations for acquiring high-quality water surface temperatures, such as split scenes for large lakes, single-channel algorithm uncertainty before Landsat 8, and lack of nighttime values for calculating daily average surface temperature (Cristóbal et al., 2009). Thus, the Zhao and Gao (2019) algorithm is a better fit for meeting the needs of this study.

Reference

- Assouline, S., Li, D., Tyler, S., Tanny, J., Cohen, S., Bou- Zeid, E., ... & Katul, G. G. (2016). On the variability of the Priestley- Taylor coefficient over water bodies. *Water Resources Research*, 52(1), 150-163.
- Cristóbal, J., Jiménez- Muñoz, J. C., Sobrino, J. A., Ninyerola, M., & Pons, X. (2009). Improvements in land surface temperature retrieval from the Landsat series thermal band using water vapor and air temperature. *Journal of Geophysical Research: Atmospheres*, 114(D8).
- Dingman, S. L. (2015). *Physical hydrology*. Waveland Press.
- Eichinger, W. E., Parlange, M. B., & Stricker, H. (1996). On the concept of equilibrium evaporation and the value of the Priestley- Taylor coefficient. *Water Resources Research*, 32(1), 161-164.
- Hostetler, S. W., & Bartlein, P. J. (1990). Simulation of lake evaporation with application to modeling lake level variations of Harney- Malheur Lake, Oregon. *Water Resources Research*, 26(10), 2603-2612.
- Li, Y., Zhao, G., Shah, D., Zhao, M., Sarkar, S., Devadiga, S., ... & Gao, H. (2021). NASA's MODIS/VIIRS Global Water Reservoir Product Suite from Moderate Resolution Remote Sensing Data. *Remote Sensing*, 13(4), 565.
- Mironov, D., Heise, E., Kourzeneva, E., Ritter, B., Schneider, N., & Terzhevik, A. (2010). Implementation of the lake parameterisation scheme FLake into the numerical weather prediction model COSMO.
- Priestley, C. H. B., & Taylor, R. J. (1972). On the assessment of surface heat flux and evaporation using large-scale parameters. *Monthly weather review*, 100(2), 81-92.
- Subin, Z. M., Riley, W. J., & Mironov, D. (2012). An improved lake model for climate simulations: Model structure, evaluation, and sensitivity analyses in CESM. *Journal of Advances in Modeling Earth Systems*, 4(1).
- Yang, Y., & Roderick, M. L. (2019). Radiation, surface temperature and evaporation over wet surfaces. *Quarterly Journal of the Royal Meteorological Society*, 145(720), 1118-1129.
- Zhao, G., & Gao, H. (2019). Estimating reservoir evaporation losses for the United States: Fusing remote sensing and modeling approaches. *Remote Sensing of Environment*, 226, 109-124.
- Zhao, G., Gao, H., & Cai, X. (2020). Estimating lake temperature profile and evaporation losses by leveraging MODIS LST data. *Remote Sensing of Environment*, 251, 112104.

Minor comments:

L69: RE ice duration: have you tried satellite-based ice phenology?

R1C8: We thank the reviewer for suggesting satellite-based ice observation, which is also brought up the reviewer 2 (R2C13). Here we explain why we didn't rely on satellite observed ice. Currently, there are no robust methods to generate reliable lake ice time series on a global scale using remote sensing data. There are several major reasons (Hall and Riggs, 2007) including 1) tradeoff of spatial and temporal resolutions of different sensors for detecting lake ice, 2) cloud contamination of satellite imageries, 3) reflectance similarity of cloud and ice/snow, and 4) difficulties to differentiate land-snow and lake-ice. Specifically, the 16-day temporal resolution

for Landsat was not appropriate to detect the freeze-up and break-up dates for lakes. Meanwhile, MODIS snow/ice product has a coarse resolution (e.g., 500 m for MOD10A2), which cannot resolve the small lakes from the HydroLAKES dataset (1.24 million lakes have an area less than 1 km²).

Even for relatively large lakes, lake ice detection using MODIS is hindered by the above reasons. For example, based on the MOD10A2 (8-day snow/ice product), the ice fraction for Lake Oneida, USA (207 km²) from MODIS shows severe data coherence issues when compared against the observed lake ice phenology (Fig. R1C8). To this end, the best practice is to use air temperature from reanalysis datasets to further correct the time series (Zhang et al., 2019). However, such correction then relies heavily on the data quality of reanalysis air temperature, leading to a similar performance with modeled ice phenology using air temperature. We have explained the above in the revised manuscript (Lines 391-397).

Fig. R1C8. Comparison of in-situ observed (Benson et al., 2012), MODIS (MOD10A2), and simulated (this study) lake ice phenology for Lake Oneida, USA.

Reference

- Hall, D. K., & Riggs, G. A. (2007). Accuracy assessment of the MODIS snow products. *Hydrological Processes: An International Journal*, 21(12), 1534-1547.
- Zhang, S., & Pavelsky, T. M. (2019). Remote sensing of lake ice phenology across a range of lakes sizes, ME, USA. *Remote Sensing*, 11(14), 1718.
- Benson, B. J., Magnuson, J. J., Jensen, O. P., Card, V. M., Hodgkins, G., Korhonen, J., ... & Granin, N. G. (2012). Extreme events, trends, and variability in Northern Hemisphere lake-ice phenology (1855–2005). *Climatic Change*, 112(2), 299-323.

L149: How did you differentiate man-made reservoirs and natural water bodies? Satellites can isolate water pixels from land pixels but they cannot differentiate water types.

R1C9: The “natural lakes” and “human-made reservoirs” are explicitly differentiated in the HydroLAKES dataset (Fig. R1C9; Messenger et al., 2016). The reservoir data in HydroLAKES are based on the Global Reservoir and Dam (GRaND) database, which identified artificial reservoirs based on data collection from multiple reservoir management and academic institutions such as U.S. Army Corps of Engineers, European Environment Agency, etc (Lehner

et al., 2011). In the HydroLAKES dataset, there are 6715 artificial reservoirs (excluding regulated natural lakes such as Lake Ontario). We directly extracted the lake/reservoir information from HydroLAKES. We have added such information in the new manuscript: “Among these lakes, HydroLAKES identified 6715 artificial lakes (i.e., reservoirs) based on the Global Reservoir and Dam database (GRanD) (Lehner et al., 2011).” (Lines 213-214).

Fig. R1C9. Shapefiles of natural lakes and artificial reservoirs from the HydroLAKES dataset.

Reference

- Messenger, M. L., Lehner, B., Grill, G., Nedeva, I., & Schmitt, O. (2016). Estimating the volume and age of water stored in global lakes using a geo-statistical approach. *Nature Communications*, 7(1), 1-11.
- Lehner, B., Liermann, C. R., Revenga, C., Vörösmarty, C., Fekete, B., Crouzet, P., ... & Wisser, D. (2011). High-resolution mapping of the world's reservoirs and dams for sustainable river-flow management. *Frontiers in Ecology and the Environment*, 9(9), 494-502.

L162: “non-precipitating forcings”: I am not sure if this is an accepted technical jargon

R1C10: Thank you for the comment. We used the term “non-precipitation forcings” to refer to meteorological forcings other than the precipitation according to North American Land Data Assimilation System (NLDAS) dataset (<https://ldas.gsfc.nasa.gov/nldas/v2/forcing>).

Eq 2: units are not consistent

R1C11: Thank you. We have double-checked the units.

Supplementary Figure 1: It would be interesting to compare your rate estimate (spatial pattern, zonal mean) with those published by other researchers.

R1C12: Thank you for the suggestion. Two types of comparisons have been conducted: 1) comparison of the zonal mean (i.e., latitudinal distribution) with Wang et al. (2018) (Fig. R1C12-1); and 2) comparison of spatial pattern with land evapotranspiration maps (Supplementary Fig. 6). Overall, our latitudinal distribution (panel **b** in Fig. R1C12-1) is similar to the results from Wang et al. (2018; panel **c**) although our curve is smoother due to higher coverage of global lakes. Meanwhile, it is worth noting that our dataset contains the evaporation rate for every single lake, and it is affected by lake-level characteristics (e.g., lake depth and fetch). Thus, it is not directly comparable to a gridded (e.g., 1 degree) lake evaporation dataset (e.g., Wang et al., 2018).

For spatial patterns, we compared our evaporation rate with land surface potential and actual evapotranspiration from multiple sources (Supplementary Fig. 6, attached here as Fig. R1C12-2). Such comparison shows the spatial consistency between our lake evaporation with potential evapotranspiration from both MODIS and Trabucco and Zomer (2019). Such information has been added in the new manuscript: “Comparisons of lake evaporation rate with actual and potential evapotranspiration over land (Supplementary Fig. 6) show a high degree of spatial consistency. The patterns of the evaporation rate zonal mean (i.e., latitudinal distribution) are also similar to the results from Wang et al. (2018) (not shown).” (Lines 281-284).

Fig. R1C12-1. Comparison of the latitudinal distribution of evaporation rate with Wang et al. (2018). Panel **a** and **b** are results from our work and panel **c** is adopted from Fig. 2b from Wang et al. (2018).

Fig. R1C12-2. Long-term average evaporation rate for a) open water (this study), b) actual evapotranspiration (from MOD16A2), c) potential evapotranspiration (from MOD16A2), and d) potential evapotranspiration (Trabucco and Zomer, 2019).

Reference

- Wang, W., Lee, X., Xiao, W., Liu, S., Schultz, N., Wang, Y., ... & Zhao, L. (2018). Global lake evaporation accelerated by changes in surface energy allocation in a warmer climate. *Nature Geoscience*, 11(6), 410-414.
- Trabucco, A., & Zomer, R. J. (2019). Global aridity index and potential evapotranspiration (ET₀) climate database v2. *CGIAR Consort Spat Inf*, 10.

Reviewer #2 (Remarks to the Author):

Review of Evaporative water loss of 1.42 million global lakes by Zhao et al.

The idea of this study is to estimate evaporation volume of global lakes from 1985 to 2018, which is ambitious. First, a global lake data from 1985 to 2018 is necessary. At present, 1.42 million global lakes from HydroLakes are produced based on STRM DEM in February, 2000, there is no available a long-term global lake dataset. The authors used global waterbody products, which is greatly different from lakes, especially impossible to acquire at monthly scale. The title of 1.42 million global lakes could be misled. In addition, monthly ice duration modeled using air temperature had large uncertainties. The ratio of lake evaporation water loss to global terrestrial evapotranspiration is not comparable as different data and spatial resolution. In addition, the writing of this manuscript is not readable, which is not followed by NC style. Overall, the data used can not support for estimates of monthly global lake evaporation changes from 1985 to 2018.

R2C0: We greatly appreciate the time and effort the reviewer had put in reviewing our manuscript! And thank you for recognizing the importance of our work. We have made substantial changes in the new manuscript and comply with the style of Nature Communications. By addressing the comments, we feel the manuscript has been much improved. In addition to our point-to-point response for each of the comments, we have summarized/clarified the major points as follows:

- 1) We have explained how we combined the dynamic water maps (GSWD; Pekel et al., 2016) with static global lake shapefiles (HydroLAKES; Messenger et al., 2016) to obtain the time series of monthly surface area for each of the 1.42 million lakes after applying an image enhancement algorithm we developed (Zhao and Gao, 2018). This algorithm can significantly reduce the impact of cloud/shadow contamination of Landsat images on the quality of area time series. By multiplying the monthly evaporation rate with monthly surface area, the monthly volumetric evaporation for each lake can be calculated from 1985 to 2018.
- 2) To better evaluate the performance of our reconstructed monthly area for the 1.42 million lakes, we have conducted more validations at the individual-lake level by comparing area estimations against in-situ observed reservoir storage/elevation time series (R2C1, Supplementary Fig. 8 and Supplementary Fig. 9).
- 3) Following the suggestion of the reviewer, we have also provided more validation of the monthly ice duration using long-term (>20 years) in-situ ice phenology data for 14 lakes (R2C2, Supplementary Fig. 10).
- 4) We have added explanation of our comparison between lake evaporation and terrestrial evapotranspiration. Because both are based on volumetric values, the comparison is not limited by the spatiotemporal resolution from either of them.
- 5) We have followed the NC style by dividing our “Results” section into topical subheadings and grouping all discussion into the new “Discussion” section and all methodology-related text has been moved to the “Methods” section. Also, we have substantially revised these sections to improve the overall quality and readability of our manuscript.

Reference

- Messenger, M. L., Lehner, B., Grill, G., Nedeva, I., & Schmitt, O. (2016). Estimating the volume and age of water stored in global lakes using a geo-statistical approach. *Nature communications*, 7(1), 1-11.
- Pekel, J. F., Cottam, A., Gorelick, N., & Belward, A. S. (2016). High-resolution mapping of global surface water and its long-term changes. *Nature*, 540(7633), 418-422.
- Zhao, G., & Gao, H. (2018). Automatic correction of contaminated images for assessment of reservoir surface area dynamics. *Geophysical Research Letters*, 45(12), 6092-6099.

Major comments:

1. In this study, the authors estimated the monthly evaporative water loss for 1.42 million lakes from 1985 to 2018. Some concepts such as “monthly” and “lakes” are not right. The global waterbody products are used directly, although it is produced at monthly scale, the gaps are large as no available data because of cloud contamination of Landsat images. In addition, the global water products are different from lakes. The authors claimed a “1.42 million lakes”, I understand a global lake mask produced in 2000 using SRTM DEM water body product (<https://doi.org/10.1038/ncomms13603>) was used. However, lakes are different from waterbody products, it is not possible to produce a global lake product, especially at monthly scale. This is the core of this study as the authors examined the trends of evaporative water loss, and the novel for global lakes.

R2C1. Thank you for this comment. The differences between our lake area estimations and the other two products (i.e., GSWD and HydroLAKES) are explained as follows.

First, our algorithm for estimating water area was developed to generate high-quality continuous monthly time series for individual lakes, with a focus on eliminating the errors from missing data due to cloud contaminations in the GSWD’s “Monthly Water History” (https://developers.google.com/earth-engine/datasets/catalog/JRC_GSW1_3_MonthlyHistory). We previously developed an image enhancement algorithm to reduce the impacts of cloud and shadow contamination (Zhao and Gao, 2018; Fig. R2C1-1). We have systematically validated this enhancement algorithm and the generated dataset (i.e., Global Reservoir Surface Area Dataset; GRSAD) using in-situ storage/elevation data of reservoirs and synthetically contaminated images (Fig. 2 and 3 in Zhao and Gao, 2018).

Fig. R2C1-1. Automatic correction of a contaminated image: (a) contaminated image, (b) raw water area, (c) water occurrence, (d) clipped occurrence, (e) histogram, and (f) enhanced water area. The image used in this example is for the E.V. Spence reservoir in Texas and was acquired on 15 June 2003. **Note** that the buffered HydroLAKES dataset was used as a processing boundary.

For this study, however, such an enhancement algorithm is too computationally expensive if applied on the 1.42 million lakes (needs continuous running for 2 to 3 years on Google Earth Engine). Thus, we developed an updated algorithm that takes advantage of the high-quality “yearly” GSWD product (https://developers.google.com/earth-engine/datasets/catalog/JRC_GSW1_3_YearlyHistory) and then reconstructed the “monthly” seasonality for each year and each lake. Compared to the “monthly” GSWD, the “yearly” GSWD provides better quality (i.e., much less cloud gaps) due to the annual image composite process. The monthly seasonality for each year was then reconstructed using the long-term climatological image for each month (January to December) (see more details in “**Lake surface area dynamics**” in the new Methods section). Such reconstructed monthly values are further validated using our previously developed dataset (GRSAD based on Zhao and Gao, 2018; Extended Data Fig. 2a). In addition, following the suggestion by the reviewer in R2C12, we have provided error statistics for each lake in GRSAD (new Extended Data Fig. 2b, 2c). These validation results against GRSAD show that our reconstructed “monthly” area values have comparable data quality as the GRSAD monthly area values.

To further validate whether such reconstructed monthly area values can capture the correct water level dynamics of lakes, we provided more validations as follows:

1. Validation against in-situ storage/elevation data. Here we used the same lakes from Zhao and Gao (2018) to be consistent (Fig. R2C1-2). The results show that our new results (based on reconstruction) even provide better quality than GRSAD for years that have fewer Landsat observations. For example, in the years from 1992 to 1998, Mossoul Lake, Iraq (Fig. R2C1-2c) has very limited Landsat observation. In GRSAD, such data gap has been filled using linear interpolation approach, which, however, cannot represent the actual seasonality

for these years. In the new results, such seasonality in these gapped years has been reconstructed.

Fig. R2C1-2. Validation of reconstructed monthly area (this study) using observed reservoir storage or elevation time series. The time series from GRSAD is also plotted for reference. The two values in the parenthesis represent R^2 values for 1) GRSAD area and observed storage or elevation and 2) reconstructed area and observed storage or elevation.

2. We further validated our area dynamics using satellite altimetry data for 12 large lakes (area > 1000 km²; Fig. R2C1-3). The area dynamics generally agree well with the elevation dynamics (R^2 range from 0.54 to 0.97). The locations with relatively low R^2 values are where the area-elevation relationships are likely nonlinear (Yigzaw et al., 2018).

We have added such information in the new manuscript: “The reconstructed monthly area values were further compared with in-situ observed storage/elevation time series (Supplementary Fig. 8) and satellite altimetry data where available (Supplementary Fig. 9). Compared to GRSAD, the new results perform better during years with fewer Landsat observations (e.g., 1992 to 1998 for Mossoul Lake, Iraq). Overall, the reconstructed areas yield comparable data quality as the GRSAD monthly area values and are capable to represent the area dynamics for lakes with a wide range of sizes.” (Lines 340-345).

Fig. R2C1-3. Validation of reconstructed monthly area (this study) using satellite altimetry elevation time series. The value in the parenthesis represents R^2 values for reconstructed area and altimetry elevation.

Second, the calculation of the global lake areas had leveraged the HydroLAKES dataset. As the reviewer pointed out, “lakes” and “water bodies” are different. While based on the SRTM Water Body Data (SWBD), HydroLAKES explicitly extracted the “lakes” (including both natural and artificial) and differentiated the lakes from other water bodies (e.g., rivers) (Fig. R2C1-4). In addition, HydroLAKES has provided the shapefiles for the 1.42 million individual lakes. By buffering outwards, a mask was generated for each lake within which the GSWD classifications were improved upon to generate continuous monthly time series (see response above).

Fig. R2C1-4. Comparison of SRTM Water Body Data (SWBD) and HydroLAKES dataset. Note that SWBD is a raster dataset and HydroLAKES is a vector (i.e., shapefile) dataset. HydroLAKES contains the lake and reservoir shapefiles by excluding river pixels.

Reference

- Zhao, G., & Gao, H. (2018). Automatic correction of contaminated images for assessment of reservoir surface area dynamics. *Geophysical Research Letters*, 45(12), 6092-6099.
- Pekel, J. F., Cottam, A., Gorelick, N., & Belward, A. S. (2016). High-resolution mapping of global surface water and its long-term changes. *Nature*, 540(7633), 418-422.
- Yigzaw, W., Li, H. Y., Demissie, Y., Hejazi, M. I., Leung, L. R., Voisin, N., & Payn, R. (2018). A new global storage- area- depth data set for Modeling reservoirs in land surface and earth system models. *Water Resources Research*, 54(12), 10-372.
- Messenger, M. L., Lehner, B., Grill, G., Nedeva, I., & Schmitt, O. (2016). Estimating the volume and age of water stored in global lakes using a geo-statistical approach. *Nature communications*, 7(1), 1-11.

2. The monthly fraction of ice duration was modeled using air temperature and freeze/thaw lags. Again, I want to know the reliability of monthly fraction of ice duration. For a global scale, air temperature in station sparse regions has large uncertainties. This could result in modelled fraction of ice duration with large uncertainties. Especially, the simulation was conducted at monthly scale.

R2C2: We agree with the reviewer that using air temperature and lake depth is a simplified method for calculating the fraction of ice duration, which is a complex phenomenon that is affected by lake energy balance, external forcings, and snow cover. However, we would like to emphasize that air temperature is still a good proxy and the most important driver for river or lake ice duration simulation (Yang et al., 2020 *Nature*; Sharma et al., 2019 *Nature Climate Change*; Woolway et al., 2020 *Nature Reviews Earth & Environment*). Specifically, the simple 0 °C air temperature isotherm method alone, together with lake area, can explain 80% of the ice-out timing variation (Arp et al., 2013). Such an air temperature-based method is markedly valid for lakes that have a small surface area (Smits et al., 2021; Caldwell et al., 2021). For large lakes, the heat storage in the summer and average winter temperature can notably impact the freeze-up and break-up timing. Thus, we modeled 1) the freeze-lag using lake average depth and 2) thaw-lag using averaged winter temperature (Extended Data Fig. 3).

While the simulated **annual** ice duration agrees well with observed data (Extended Data Fig. 3d), here we further validated the **monthly** ice duration using long-term in-situ ice phenology data. We collected long-term (≥ 20 years) ice phenology data for 14 lakes in the United States (Benson et al., 2012). Overall, the monthly fraction of ice duration agrees well with in-situ observed data (Fig. R2C2, also added in the manuscript as Supplementary Fig. 10). The average bias is about 7 days/year (range from -19 to 33 days/year). Note that we have quantified the impact of the simulation uncertainty of $f_{d,ice}$ on the evaporation volume uncertainty (Lines 416-423). Such monthly validation results have been added in the new manuscript: “Second, the monthly simulation performance of these two fitted equations (Extended Data Fig. 3b, 3c) was evaluated using long-term (≥ 20 years) in-situ ice phenology data for 14 North America lakes (Supplementary Fig. 10). The average bias for these lakes is 7

days/year (ranges from -19 to 33 days/year), indicating of satisfactory data quality.” (Lines 381-384).

We agree with the reviewer that air temperatures from reanalysis datasets may not be accurate for specific regions due to a limited monitoring network. That is the reason why we used 3 reanalysis datasets to reduce such uncertainty. In addition, with the help of modern data assimilation (DA) algorithms and increasingly accessible satellite data, the data quality and coherence for reanalysis datasets have been significantly improved in recent decades. For example, the 4D-Var DA system, which ERA5 is based on, assimilated a large volume of satellite data in recent decades with an atmospheric sounder to infer the atmospheric temperature (Uppala et al., 2005; Hersbach et al., 2020). Such reanalysis datasets provide our best understanding of the global climatic system. Thus, regions with a limited number of ground stations can still have satisfactory performance in terms of air temperature. For instance, the mean bias of ERA-Interim (early version of ERA5) in Tibet Plateau is -0.6°C and RMSE is 1.3°C (Zou et al., 2014). We have added such discussion in the new manuscript: “The air temperature uncertainty of reanalysis datasets is originated from multiple sources including data assimilation algorithm and assimilated data sources (i.e., satellites and ground stations).” (Lines 418-421).

Fig. R2C2. Validation of monthly fraction of ice duration using long-term in-situ ice phenology data.

References

- Yang, X., Pavelsky, T. M., & Allen, G. H. (2020). The past and future of global river ice. *Nature*, 577(7788), 69-73.
- Sharma, S., Blagrove, K., Magnuson, J. J., O'Reilly, C. M., Oliver, S., Batt, R. D., ... & Woolway, R. I. (2019). Widespread loss of lake ice around the Northern Hemisphere in a warming world. *Nature Climate Change*, 9(3), 227-231.
- Woolway, R. I., Kraemer, B. M., Lenters, J. D., Merchant, C. J., O'Reilly, C. M., & Sharma, S. (2020). Global lake responses to climate change. *Nature Reviews Earth & Environment*, 1(8), 388-403.
- Arp, C. D., Jones, B. M., & Grosse, G. (2013). Recent lake ice- out phenology within and among lake districts of Alaska, USA. *Limnology and Oceanography*, 58(6), 2013-2028.
- Smits, A. P., Gomez, N. W., Dozier, J., & Sadro, S. (2021). Winter Climate and Lake Morphology Control Ice Phenology and Under- Ice Temperature and Oxygen Regimes in Mountain Lakes. *Journal of Geophysical Research: Biogeosciences*, 126(8), e2021JG006277.
- Caldwell, T. J., Chandra, S., Albright, T. P., Harpold, A. A., Dilts, T. E., Greenberg, J. A., ... & Dettinger, M. D. (2021). Drivers and projections of ice phenology in mountain lakes in the western United States. *Limnology and Oceanography*, 66(3), 995-1008.
- Uppala, S. M., Kållberg, P. W., Simmons, A. J., Andrae, U., Bechtold, V. D. C., Fiorino, M., ... & Woollen, J. (2005). The ERA- 40 re- analysis. *Quarterly Journal of the Royal Meteorological Society: A journal of the atmospheric sciences, applied meteorology and physical oceanography*, 131(612), 2961-3012.
- Hersbach, H., Bell, B., Berrisford, P., Hirahara, S., Horányi, A., Muñoz- Sabater, J., ... & Thépaut, J. N. (2020). The ERA5 global reanalysis. *Quarterly Journal of the Royal Meteorological Society*, 146(730), 1999-2049.
- Benson, B. J., Magnuson, J. J., Jensen, O. P., Card, V. M., Hodgkins, G., Korhonen, J., ... & Granin, N. G. (2012). Extreme events, trends, and variability in Northern Hemisphere lake-ice phenology (1855–2005). *Climatic Change*, 112(2), 299-323.
- Zou, H., Zhu, J., Zhou, L., Li, P., & Ma, S. (2014). Validation and application of reanalysis temperature data over the Tibetan Plateau. *Journal of Meteorological Research*, 28(1), 139-149.

3. The writing of this manuscript is not readable. It is okay for Abstract and Introduction from Line 40-62. However, after Line 63, too many numbers and formulars. The structure of this manuscript did not flow the style of Nature Communications, and need rewrite. Especially, the results and discussion should be separated from methods clearly.

R2C3: Thank you for pointing out the style of Nature Communications. In the new manuscript, we have divided our “Results” section into topical subheadings including “Spatial heterogeneity of lake evaporation volume”, “Long-term trends of lake evaporation volume”, and “Attribution of trends and variability of lake evaporation volume”. We have also grouped all discussions into the new “Discussion” section and all methodology-related text has been moved to the “Methods” section.

We'd also like to thank your suggestion in improving the readability of this manuscript. We have considered your suggestion and comments very seriously. By clearly separating results and discussion from the methods, the equations are now centralized in the method section and the data in results. We feel the readability has been much improved.

We agree with the reviewer that many formulas and numbers had been used. We did put a lot of thoughts into it with different scenarios in revisions but ended up choosing to keep most of the numbers and formulas. This decision is based on two considerations. First, it will better serve the objective of this work, which is to provide the first quantitative estimates of evaporation volume from global lakes and to evaluate the long-term trend. The analysis of the evaporation

volume involves evaporation rate, lake surface area, open water area, and the fraction of ice duration. We could not figure out an alternative/better way to describe these quantities and changes in a more effective and concise (also most intuitive) manner other than relying on numbers and formulas. Such use of numbers and formulas is also consistent with other global lake studies (e.g., Cooley et al., Nature 2021; Jane et al., Nature 2021; Yang et al., Nature 2020). The other two reviewers did not raise a similar concern. In fact, Reviewer #3 states “*The methods are well described and results well communicated.*” By keeping the numbers and formulas mostly in the method section and reorganizing the manuscript, we believe the manuscript flows much better now.

Reference

- Cooley, S. W., Ryan, J. C., & Smith, L. C. (2021). Human alteration of global surface water storage variability. *Nature*, 591(7848), 78-81.
- Jane, S. F., Hansen, G. J., Kraemer, B. M., Leavitt, P. R., Mincer, J. L., North, R. L., ... & Rose, K. C. (2021). Widespread deoxygenation of temperate lakes. *Nature*, 594(7861), 66-70.
- Yang, X., Pavelsky, T. M., & Allen, G. H. (2020). The past and future of global river ice. *Nature*, 577(7788), 69-73.

4. The analysis of increasing evaporation linking with lake ice coverage and lake area is simple. It is not a quantitative evaluation, and only calculated by turning off one of variables.

R2C4: Thanks for the comment. The increasing trends of volumetric evaporation are attributed to three factors: evaporation rate, lake surface area, and lake ice duration. To evaluate how the trend of evaporation volume is attributed to the trends of these three components, we used a simple detrending method. We removed the trend of each component for each lake while keeping the other two components unchanged. Despite its simplicity, such a method was able to provide a quantitative evaluation of how the trend of each component contributes to the total trend. Because of its simplicity and capability in attribution detection, such a method has been broadly used for attribution studies in climate and ecology communities (Ballard et al., 2019; Zhang et al., 2016; Wang et al., 2018).

Reference

- Ballard, T. C., Sinha, E., & Michalak, A. M. (2019). Long-term changes in precipitation and temperature have already impacted nitrogen loading. *Environmental Science & Technology*, 53(9), 5080-5090.
- Zhang, J., Sun, F., Xu, J., Chen, Y., Sang, Y. F., & Liu, C. (2016). Dependence of trends in and sensitivity of drought over China (1961–2013) on potential evaporation model. *Geophysical Research Letters*, 43(1), 206-213.
- Wang, T., Sun, F., Xia, J., Liu, W., Sang, Y., & Wang, H. (2018). An experimental detrending approach to attributing change of pan evaporation in comparison with the traditional partial differential method. *Journal of Hydrology*, 564, 501-508.

Specific comments:

- “reservoirs” is manmade lakes? How to differentiate lakes from reservoirs?

R2C5: Thank you for the questions. By definition, a reservoir is “*a large natural or artificial lake used as a source of water supply*”. More specifically, the HydroLAKES dataset differentiates 3 types of lakes: 1) natural lakes, 2) regulated natural lakes, and 3) artificial reservoirs. We followed such a definition by referring “reservoirs” as the third type (i.e., artificial/manmade lakes). To avoid confusion, we have replaced the word “manmade” with “artificial” in the new manuscript.

The “natural lakes” (type 1 and 2) and “human-made reservoirs” (type 3) are explicitly differentiated in HydroLAKES (Fig. R2C5; Messenger et al., 2016). The reservoir data in HydroLAKES are based on the Global Reservoir and Dam (GRanD) database, which identified artificial reservoirs based on data collection from multiple reservoir management and academic institutions such as the U.S. Army Corps of Engineers, European Environment Agency, etc (Lehner et al., 2011).

We have added such information in the new manuscript: “Among these lakes, HydroLAKES identified 6715 artificial lakes (i.e., reservoirs) based on the Global Reservoir and Dam database (GRanD) (Lehner et al., 2011).” (Lines 213-214).

Fig. R2C5. Locations of the natural lakes and artificial reservoirs across the world according to the HydroLAKES dataset (Messenger et al., 2016).

Reference

- Messenger, M. L., Lehner, B., Grill, G., Nedeva, I., & Schmitt, O. (2016). Estimating the volume and age of water stored in global lakes using a geo-statistical approach. *Nature communications*, 7(1), 1-11.
- Lehner, B., Liermann, C. R., Revenga, C., Vörösmarty, C., Fekete, B., Crouzet, P., ... & Wisser, D. (2011). High-resolution mapping of the world's reservoirs and dams for sustainable river-flow management. *Frontiers in Ecology and the Environment*, 9(9), 494-502.

- “The monthly lake surface area was reconstructed using a Landsat-based global surface water dataset¹⁷” The authors did not understand the global waterbody product well, it cannot be used as a global lake product or constructed product, especially at a monthly scale.

R2C6: We agree with the reviewer that the Landsat-based global surface water database (GSWD; Pekel et al., 2016) cannot be used as a global lake product directly. That is why we derived our own lake area product by combining the GSWD and SRTM-based global lake shapefile dataset (i.e., HydroLAKES; Messenger et al., 2016). The former dataset represents **dynamic** water changes, and the latter is **static** lake shapefile, which is used as the mask to extract the area time series from the former dataset.

The reconstructed monthly area dataset has significantly reduced the impact of data gaps due to cloud contamination of Landsat images. Following the suggestion of the reviewer, we have provided more validations at single lake levels using in-situ storage/elevation values and altimetry elevation values (for more details please refer to R2C1). Together with the cross-validation at the global scale using GRSAD (Extended Data Fig. 2), the results show that such reconstructed dataset can (1) provide comparable (or even better for years that have limited Landsat observations) quality with our previously developed high-quality dataset (i.e., GRSAD; Zhao and Gao, 2018) and (2) can represent the water level/storage dynamics at single lake level for a wide range of lake sizes.

Reference

- Zhao, G., & Gao, H. (2018). Automatic correction of contaminated images for assessment of reservoir surface area dynamics. *Geophysical Research Letters*, *45*(12), 6092-6099.
- Pekel, J. F., Cottam, A., Gorelick, N., & Belward, A. S. (2016). High-resolution mapping of global surface water and its long-term changes. *Nature*, *540*(7633), 418-422.
- Messenger, M. L., Lehner, B., Grill, G., Nedeva, I., & Schmitt, O. (2016). Estimating the volume and age of water stored in global lakes using a geo-statistical approach. *Nature communications*, *7*(1), 1-11.

- “This value is 15.4% higher than previous estimates...” How you concluded that your value is higher than previous study. It is compared in same study period and condition?

R2C7: Thanks for the question. To our best knowledge, there is no estimation of global volumetric evaporation by considering both evaporation rate and lake area dynamics. Thus, we were not able to compare our work unequivocally with any previous estimation. To this end, we still would like to put our estimation into context and thus we used the well-cited value 1300 km³/year from Oki and Kanae (2006 Science), which provides approximate estimates for global lakes based on global hydrological model.

Reference

- Oki, T., & Kanae, S. (2006). Global hydrological cycles and world water resources. *science*, *313*(5790), 1068-1072.

- “To quantify the VE/ET ratio, we calculated the land ET using the MODIS...” MODIS data has a coarse resolution relative to 30-m Landsat water body products, and moreover, MODIS data is

available after 2000, but Landsat water product used is available from 1985. However, you calculated the ratio of V_E/ET ratio, it is not reasonable.

R2C8: We thank the reviewer for the comment. Because the lake V_E and land ET were both calculated as volumetric values (e.g., $1500 \text{ km}^3/\text{year}$), the comparison is independent of the spatial or temporal resolution of both products. Such information has been corrected in the new manuscript: “While the total lake surface area only accounts for 1.57% of the global land area, lake evaporation plays an important role in global and regional terrestrial evapotranspiration (ET), as manifested through the V_E/ET ratio. The land ET volume was quantified by multiplying the Moderate Resolution Imaging Spectroradiometer (MODIS) Global Terrestrial Evapotranspiration product (i.e., MOD16A2) to the land surface area, and then the total ET volume was derived by adding the lake evaporation volume to the land ET volume.” (Lines 102-107). In addition, because MODIS data started from Mar 2000 (as mentioned by the reviewer) and our Landsat-based product started from 1985, we selected the overlapping period (i.e., 2001 to 2018) as the basis of the comparison. Such information can be found in the caption of Fig. 2 in the new manuscript: “Fig. 2. Percentage of lake evaporation versus total land evapotranspiration (V_E/ET) over the period from 2001 to 2018 for each HydroSHEDS level-3 basin.” (Lines 115-116).

- How to understand $1.57 \text{ mm}/\text{yr}/\text{yr}$?

R2C9: This unit denotes the long-term trend of evaporation rate. The annual evaporation rate is in the unit of mm/year . Thus, the slope of the annual evaporation rate is $\text{mm}/\text{year}/\text{year}$, which is commonly used to denote the trend of evaporation (Liu et al., 2018; Stephens et al., 2018).

Reference:

- Liu, M., Adam, J. C., Richey, A. S., Zhu, Z., & Myneni, R. B. (2018). Factors controlling changes in evapotranspiration, runoff, and soil moisture over the conterminous US: Accounting for vegetation dynamics. *Journal of hydrology*, 565, 123-137.
- Stephens, C. M., McVicar, T. R., Johnson, F. M., & Marshall, L. A. (2018). Revisiting pan evaporation trends in Australia a decade on. *Geophysical Research Letters*, 45(20), 11-164.

- “Globally, the long-term trend for V_E is $2.1\%/\text{decade}$ ($3.12 \pm ??? \text{ km}^3/\text{yr}$).” The uncertainties of trend should be included. Other values are same problems.

R2C10: Thank you. The slope uncertainty mainly inherits the uncertainty of the evaporation volume. Because we have quantified the uncertainty of evaporation volume as 9.93%, we thus employed Monte Carlo simulation to randomly sample from the Gaussian distribution of evaporation volume for each year. After 1 million cycles of Monte Carlo simulations, we have quantified the slope uncertainty as $\pm 2.4 \text{ km}^3/\text{yr}$. Because we already have many numbers in the manuscript as pointed out by reviewer #2 (R2C3), and adding uncertainty for each trend value may reduce the readability of the manuscript, we have added such information into the new manuscript only for the global V_E trend values (Lines 133-134).

Fig. R2C10. The slope uncertainty based on 1 million cycles of Monte Carlo simulations.

- Reservoirs from HydroLakes is a static dataset. How you calculated reservoir evaporative loss from 1985 to 2018?

R2C11: As explained in R2C1 and in the new “Methods” section, the monthly time series for both evaporation rate and lake surface area (from **dynamic** global surface water dataset, GSWD; Pekel et al., 2016) were calculated for each of the 1.42 million lakes in HydroLAKES dataset. Here we use an example for Lake Mead, USA to show the monthly time series for evaporation rate, surface area, and the calculated evaporation volume.

Fig. R2C11. Monthly time series and seasonality of a) surface area, b) evaporation rate, and c) volumetric evaporation for Lake Mead, USA. The blue line in panel a represents the in-situ observed evaporation rate based on the Eddy Covariance method. The blue line in panel b represents the in-situ observed storage time series.

Reference

Pekel, J. F., Cottam, A., Gorelick, N., & Belward, A. S. (2016). High-resolution mapping of global surface water and its long-term changes. *Nature*, 540(7633), 418-422.

- Validation of reconstructed monthly surface area by compared to 6715 reservoirs. The high R2 does not mean high quality, which is determined by number of samples. More importantly, the mapped reservoir areas for comparison are from same date?

R2C12: Thank you. We agree with the reviewer that R^2 only represents the overall agreement of the scattered cloud. Thus, we further calculated the relative bias (rBias) and the relative Root Mean Square Error (rRMSE) for each lake (Fig. R2C12, added in the manuscript as Extended Data Fig. 2). Overall, the median rRMSE value is 9.2% and median rBias is 1.6%, showing the good performance of the reconstructed monthly area. The reviewer was correct that such comparisons are only for the matching months (e.g., June 2018 from the reconstructed area vs. June 2018 from GRSAD). We have added the new evaluation to the revised manuscript: “In addition, we calculated the error statistics for each of the 6715 reservoirs (Extended Data Fig. 2b, 2c). The median relative bias (rBias) of 1.6% and median relative Root Mean Square Error (rRMSE) of 9.2% further manifest the high quality of the reconstructed monthly area values.” (Lines 336-339).

Fig. R2C12. Validation of reconstructed monthly lake area using the GRSAD monthly clear (contamination percentage $\leq 5\%$) values.

- The simulation of lake ice phenology is simple. Actually, lake ice phenology can be observed from MODIS data. However, MODIS has a coarse resolution, which is incomparable to Landsat waterbody data used in this study.

R2C13: Thank you for suggesting the MODIS snow/ice product. Please refer to R2C2 for the validity of using air temperature and freeze/thaw lags to simulate the lake ice phenology. Here, we would like to add the comparison of air temperature-based simulation and MODIS-based ice phenology using Lake Oneida, USA as an example (Fig. R2C13). Due to cloud contamination and omission/commission errors in MODIS snow/ice product (MOD10A2), the time series of MODIS-based ice phenology has severe data coherence issues. On the other hand, the air temperature-based ice phenology agrees well with the in-situ observed values on a monthly scale (bias is -2 days/year).

Fig. R2C13. Comparison of in-situ observed (Benson et al., 2012), MODIS (MOD10A2), and simulated lake ice phenology for Oneida Lake, USA.

In summary, we did not rely on the satellite-based ice phenology observations for three reasons. First, cloud contamination of satellite images and reflectance similarity of cloud and ice/snow can notably limit the detection coverage and accuracy of lake ice (Hall and Riggs, 2007). Second, it is difficult to differentiate ice/snow-covered lake and land without knowing the precise boundary/shoreline of lakes. Third, the tradeoff of spatial and temporal resolution of various satellite sensors also limited their capability to provide a globally consistent dataset. For high-resolution sensors, they typically have a coarse temporal resolution (e.g., 16-day for Landsat), thus not appropriate to detect the freeze-up and break-up dates for lakes. Meanwhile, MODIS sensors have a coarse resolution. For example, the Normalized Difference Snow Index (NDSI) product (i.e., MOD10A1 and MOD10A2) has a resolution of 500 meters. Thus, such a product cannot resolve the small lakes from the HydroLAKES dataset (1.24 million lakes have an area less than 1 km²). Such information has been added in the new manuscript: “We did not use remotely sensed ice cover (e.g., MODIS) to represent the ice phenology for two reasons: 1) the tradeoff between spatial and temporal resolution significantly limit the wide application of satellite ice observations, especially for small lakes; 2) cloud cover and reflectance similarity between cloud and surface ice/snow reduce the data coherence of ice phenology time series (Supplementary Fig. 12). To correct the data coherence issue, reanalysis air temperature is commonly used to smooth the ice cover time series from satellite data. However, such a method then relies on the data quality of reanalysis air temperature, leading to a similar performance with our modeling approach (Zhang and Pavelsky, 2019).” (Lines 391-397).

Reference

- Hall, D. K., & Riggs, G. A. (2007). Accuracy assessment of the MODIS snow products. *Hydrological Processes: An International Journal*, 21(12), 1534-1547.
- Zhang, S., & Pavelsky, T. M. (2019). Remote sensing of lake ice phenology across a range of lakes sizes, ME, USA. *Remote Sensing*, 11(14), 1718.
- Benson, B. J., Magnuson, J. J., Jensen, O. P., Card, V. M., Hodgkins, G., Korhonen, J., ... & Granin, N. G. (2012). Extreme events, trends, and variability in Northern Hemisphere lake-ice phenology (1855–2005). *Climatic Change*, 112(2), 299-323.

Reviewer #3 (Remarks to the Author):

This manuscript is tremendous effort that provides the first volumetric estimate of evaporation for global lakes while also generating an interesting global dataset. I believe the focus on water volume as opposed to rate is a valuable and original contribution. The methods are well described and results well communicated.

I have few suggestions to improve this manuscript. My suggestions mostly involve clarifying some writing in the abstract/introduction and 1 other paragraph. Also, there are a lot of results and information in this manuscript, therefore I encourage the authors to reduce some of the results if possible (but only if there is something they believe is less important for the paper). Detailed comments are below.

R3C0: Dear Dr. Gardner, we sincerely appreciate your time and effort in reviewing our manuscript. Also, thanks for pointing out the scientific contribution of our work. Admittedly, there is a lot of information in our manuscript due to the global coverage and the large number of lakes studied. We have followed your suggestions by clarifying the writing and reducing some results. Specifically, we have grouped the results into three topical subheadings: “**Spatial heterogeneity of lake evaporation volume**”, “**Long-term trends of lake evaporation volume**”, and “**Attributions of trends and variability of lake evaporation volume**” to improve the readability of all the results. We hope that the revised and reorganized manuscript can better deliver the message of our work to the general audience.

Abstract

Generally, I recommend using more clear language to articulate why this is important. Simply saying we don't know the distributions (and of what?) and that it is intrinsically complex is a bit vague. I think the volume aspect is a better “hook” for this paper, at least in the abstract. Stating simply, we don't know the volume of water lost from lakes because we focus on rates. And explain in 1 sentence why evaporative volume matters.

R3C1: Thank you. Yes, the “volume” aspect of evaporation is one of the major contributions of our work. Previous studies mostly focused on reservoir/lake evaporation rate and typically ignored the lake area dynamics. However, evaporation volume estimation from lakes is important for lake water/energy budget quantification, moisture recycling and vapor transfer modeling, and water resources management (especially in the arid regions). In the new abstract, we have removed the vague statements such as “intrinsically complex” and “distribution”, and just simply stated that “**However, the evaporative volume from global lakes – from the spatial distribution to long-term trend – is as of yet unknown**” (Lines 22-23) following the importance of evaporative water losses in the first sentence.

Line 22: delete “human”.

R3C2: Done.

Line 22 and 47: what is meant by distributions (of evaporative water loss)? Spatial variability or distribution, the distribution/histogram of all values of lake evaporation? Please clarify this word or remove to make more simple.

R3C3: Thanks. In the previous version, we meant spatial heterogeneity by using the word “distribution”. We have clarified this by saying “**volume from global lakes – from the spatial distribution to long-term trend – is as of yet unknown**”.

Line 23: What is intrinsic complexity? Please use simple language for the abstract or consider not using this word intrinsic complexity.

R3C4: Thank you. We have removed such wording. Previously, we were trying to say that quantification of evaporative loss from lakes on a global scale is difficult due to the complexity of lake system modeling and monitoring.

Main

Line 42: artificial lakes support ecosystems as well. Perhaps keep it to a more general word like... “Lakes support aquatic and terrestrial biodiversity and are an important water resource for humans”

R3C5: Thank you. Yes, reservoirs can also have the functional purpose of ecosystem services. We have edited this sentence: “**Lakes support aquatic and terrestrial biodiversity and are an important water resource for humans (Domisch et al., 2015; Wisser et al., 2010).**” (Lines 41-42)

Reference

- Domisch, S., Amatulli, G., & Jetz, W. (2015). Near-global freshwater-specific environmental variables for biodiversity analyses in 1 km resolution. *Scientific data*, 2(1), 1-13.
- Wisser, D., Frolking, S., Douglas, E. M., Fekete, B. M., Schumann, A. H., & Vörösmarty, C. J. (2010). The significance of local water resources captured in small reservoirs for crop production—A global-scale analysis. *Journal of Hydrology*, 384(3-4), 264-275.

Line 46: Consider taking a sentence or two to explain how the climate change impacts in parentheses impact evaporative loss rather than just listing them. Particularly, consider explaining the Bowen ratio in plain language.

R3C6: Thank you. We have replaced the parentheses with a new sentence to explain which processes might change lake areas and evaporation rate “**For instance, evaporation rate can be altered by warming temperature (O'Reilly et al, 2015; Wang et al, 2018) and elevated solar radiation (Zhao and Gao, 2019), while open water area can increase from shrinking lake ice cover (Sharma et al., 2019) or decrease from extreme drought conditions (Van Dijk et al., 2013)**”. Because the Bowen ratio is the ratio of sensible heat over latent heat and may not be appealing to the general audience in this first paragraph, we have removed such language.

Reference

- Sharma, S., Blagrove, K., Magnuson, J. J., O'Reilly, C. M., Oliver, S., Batt, R. D., ... & Woolway, R. I. (2019). Widespread loss of lake ice around the Northern Hemisphere in a warming world. *Nature Climate Change*, 9(3), 227-231.
- Van Dijk, A. I., Beck, H. E., Crosbie, R. S., de Jeu, R. A., Liu, Y. Y., Podger, G. M., ... & Viney, N. R. (2013). The Millennium Drought in southeast Australia (2001–2009): Natural and human causes and implications for water resources, ecosystems, economy, and society. *Water Resources Research*, 49(2), 1040-1057.
- O'Reilly, C. M., Sharma, S., Gray, D. K., Hampton, S. E., Read, J. S., Rowley, R. J., ... & Zhang, G. (2015). Rapid and highly variable warming of lake surface waters around the globe. *Geophysical Research Letters*, 42(24), 10-773.
- Wang, W., Lee, X., Xiao, W., Liu, S., Schultz, N., Wang, Y., ... & Zhao, L. (2018). Global lake evaporation accelerated by changes in surface energy allocation in a warmer climate. *Nature Geoscience*, 11(6), 410-414.
- Zhao, G., & Gao, H. (2019). Estimating reservoir evaporation losses for the United States: Fusing remote sensing and modeling approaches. *Remote Sensing of Environment*, 226, 109-124.

Line 48-49: I agree it is crucial, but I am not sure how understanding evaporation mitigates climate and support sustainable ecosystems. This is a great paper and analysis, and I do not think it requires vague justifications. Consider just saying (after you just explain a bit about how climate change is impacting evaporative loss through the 3 mechanisms in parentheses), "It is crucial to understand trends and drivers of evaporative water loss from lakes at a global scale." Paragraph 2 is great. I think the argument for evaporative volume taking into account the surface water and ice dynamics is the strength of this work.

R3C7: Thank you. We have removed the vague justifications of "mitigate the adverse impacts of climate change and support ecosystems" and simply state that "It is crucial to understand spatiotemporal changes and drivers of evaporative water loss from lakes at a global scale". Thank you for identifying the strength of our work in the second paragraph. We believe that the quantification of evaporative water loss (by considering both evaporation rate and lake area dynamics) can provide more information to global hydrological/climatic modeling and water resources management and thus hope our work can draw attention from multiple communities and make impacts on how we view/manage the precious water resources.

Line 144: This is very interesting! That reservoirs contribute more water loss.

R3C8: Thank you. We were also surprised that the reservoir evaporative water loss is equivalent to 20% of global annual consumptive water use, meaning that evaporation is a big but "invisible water user" of our reservoir water storage. Meanwhile, due to the rising global temperature, the role evaporation plays in reservoir water loss might be higher in the future, really pressing the need for better water resources management strategies, especially in the semi-arid and arid regions.

Line 160: Consider saying "In addition to" rather than "besides".

R3C9: Thank you. The wording has been changed.

Line 172: This doesn't seem like topic sentence for what this paragraph is about.

R3C10: Thank you. We have reorganized the related paragraphs that describe the results from Fig. 4. The topical sentence has been added: "The percentage contributions of the three factors (E_{lake} , $f_{d,ice}$, and A_s) to the inter-annual changes of lake evaporation volume have shown clear spatial patterns (Fig. 4; see Methods)." and then the two categories were separated described (Lines 159-160).

Line 173: 77% of lakes have exactly this area? Are less than this area? Or combined have this area? Please clarify this sentence.

R3C11: We apologize for the confusion. This is the combined area, which has been clarified in the new manuscript: "There are 1.09 million lakes (77%) that belong to this category with a combined total open water area of 1.13×10^6 km²" (Lines 163-164).

172-178: This paragraph needs some attention. Each sentence seems to be about a different result and topic. Please clarify this paragraph or remove results/sentences in it that are less important to the story.

R3C12: Thank you. This paragraph follows the previous paragraph to discuss the second category. We have edited both paragraphs to clarify the definition of the two categories and also made some changes to smooth the transition between sentences (Lines 159-180).

Line 206: "physically-based" model not physical-based.

R3C13: Thank you. Done.

Line 214: remove "sheds some critical light on" and change to "...and shows how lake systems are responding..."

R3C14: Thanks. We have changed the text (Lines 205-207).

John Gardner

Reviewers' Comments:

Reviewer #1:

Remarks to the Author:

I appreciate the authors' response to my questions and the efforts they made to improve clarity. I agree that the Penman model in its original form is based on first principles. But actual implementation of this model requires heavy parameterizations (e. g., the wind function, the storage term, and T_e). I am not fully convinced that by assembling parameterizations developed separately by other people and for different purposes, the Penman model can give robust prediction.

I recently became aware of a comprehensive dataset published by Zhang et al. (2020, Earth System Science Data 12, 2635-2645) for a large lake. I recommend that they do a performance evaluation against this dataset. Because this lake is larger than the original wind function was intended for, this evaluation is a good test of their modeling system. It will be helpful (1) to compare the evaporation calculated for this lake (without any additional tuning) with the observed evaporation at monthly, annual and interannual timescales, and (2) to evaluate the wind function, the storage term parameterization, and key input variables (incoming solar and incoming longwave) against the observational data. This second task will shed light on the question as to whether good agreement in evaporation rate is caused by cancelation of errors, or in the event of bad agreement, what is the main source of error.

One limitation of their model is that it does not work when the lake is covered by ice. The rising trend in evaporation can result directly from changes in "non-precipitating forcings" and indirectly via shortening of ice cover duration. Can you evaluate this confounding effect?

Reviewer #2:

Remarks to the Author:

I have three major concerns:

(1) Landsat-based global surface water database (GSWD) from Pekel et al. (2016) provides monthly frequency of water pixels mapped. If the pixels are cloud covered, the pixels will be replaced with cloud-free pixels in other time. Therefore, monthly pixels of waterbodies from GSWD are not real status of waterbody at a specific date. Pekel et al. (2016) showed this clear, and only use the waterbody frequency of more than 75% for trend examination. Especially, waterbody is not lake, many works need to do for lake extraction from waterbody. The authors used a global lake product from HydroLAKES, which is derived from SRTM DEM in February 2000 and includes 1.42 million lakes. The authors used the GSWD directly and claimed they examined evaporative water loss of 1.42 million global lakes. Obviously, the authors did not understand how the GSWD was produced, and lake mapping well, which can cause error estimate of the trend of the evaporative volume for global lakes. I do not think the authors can address this problem appropriately.

(2) The error data of lakes (concept) used can result in error conclusion such as increasing lake surface area accounts for 19% of global lake evaporative water loss.

(3) Lake ice phenology was estimated using air temperature and freeze/thaw lags. I do not think this is a reliable estimate as which has no physical explanation and air temperature has large uncertainties in station sparse regions.

Based these major considerations, I do not think the authors provide a reliable estimate of global lake evaporative water loss, and attribution analysis such as decreasing lake ice coverage and the increasing lake surface area.

Reviewer #3:

Remarks to the Author:

The authors have thoroughly addressed my comments, I and believe those of the other reviewers. I have no further edits or comments. Only that the authors should consider making their code publicly available (e.g. github etc.) if possible. I recommend this for publication.

Reviewer #1 (Remarks to the Author):

I appreciate the authors' response to my questions and the efforts they made to improve clarity. I agree that the Penman model in its original form is based on first principles. But actual implementation of this model requires heavy parameterizations (e. g., the wind function, the storage term, and T_e). I am not fully convinced that by assembling parameterizations developed separately by other people and for different purposes, the Penman model can give robust prediction.

I recently became aware of a comprehensive dataset published by Zhang et al. (2020, Earth System Science Data 12, 2635-2645) for a large lake. I recommend that they do a performance evaluation against this dataset. Because this lake is larger than the original wind function was intended for, this evaluation is a good test of their modeling system. It will be helpful (1) to compare the evaporation calculated for this lake (without any additional tuning) with the observed evaporation at monthly, annual and interannual timescales, and (2) to evaluate the wind function, the storage term parameterization, and key input variables (incoming solar and incoming longwave) against the observational data. This second task will shed light on the question as to whether good agreement in evaporation rate is caused by cancelation of errors, or in the event of bad agreement, what is the main source of error.

One limitation of their model is that it does not work when the lake is covered by ice. The rising trend in evaporation can result directly from changes in "non-precipitating forcings" and indirectly via shortening of ice cover duration. Can you evaluate this confounding effect?

We would like to thank the reviewer again for the comments. We have followed your suggestions to further validate our evaporation rate algorithm at different time scales, and for different energy terms, using the recommended observations at Lake Taihu (and additional observations at Lake Mead).

R1C1. Algorithm performance at Lake Taihu

We first compared our results from this manuscript (without any additional tuning) with the in-situ observed evaporate rate for Lake Taihu at both monthly and annual time steps (Fig. R1-1). From Zhang et al. (2020), there are 5 sites (i.e., BFG, DPK, MLW, PTS, and XLS) that provide long-term (>5 years) in-situ data. Both Fig. R1-1a and R1-1b show good agreement between observed and simulated evaporation rates. The long-term bias of the averaged simulated results from the averaged in-situ data is -0.03 mm/day (-1%), and the coefficient of determination (R^2) values are 0.98 and 0.83 for the monthly and annual scales.

Fig. R1-1. Comparison of the evaporation rates of Lake Taihu at the monthly and annual time scales. The shaded area represents the uncertainty from different input forcing datasets (i.e., TerraClimate, ERA5, and GLDAS).

R1C2. Evaluation of error cancellation

We have also calculated each of the energy terms specified in the Penman equation: net radiation (R_n), heat storage changes (δU), and latent heat flux (λE_{lake}). We have made these calculations for Lake Taihu (Fig. R1-2) as the reviewer suggested, and we have also added results for Lake Mead (Fig. R1-3). Lake Taihu has an average depth of 1.9m, while Lake Mead's average depth is greater than 50m. Therefore, such energy term comparisons can fully evaluate the robustness of our algorithm.

$$\lambda E_{lake} = \frac{\Delta(R_n - \delta U) + \gamma \lambda (a + bu_2) L_f^{-0.1} \cdot VPD}{\Delta + \gamma} \quad (R1)$$

For both lakes, the input net radiation, the simulated heat storage changes, and the simulated latent heat flux all agree well with the observed values. This has two implications: First, it directly indicates a satisfactory performance of the algorithm with regard to the simulation of the energy budget of lakes. Second, it indirectly suggests that the aerodynamic term in the Penman equation is also correctly simulated.

In addition, we have compared the reanalysis VPD and wind speed values with their in-situ counterparts. Despite the large uncertainties among the three reanalysis datasets (i.e.,

TerraClimate, ERA5, and GLDAS), all three have shown a strong temporal correlation with observation values at both lakes. For Lake Taihu (Fig. R1-2d), the R^2 value between the averaged VPD of the three reanalysis datasets and the averaged observed VPD of the five sites is 0.92. Because the aerodynamic term in the Penman equation is a function of the “wind function” and the VPD, this further suggests that the “wind function” representation in the Penman equation is correctly simulated. For actual wind speed, the reanalysis data lies between the minimum and maximum observed values at the 5 in-situ sites. Additionally, there is also a strong temporal correlation between the reanalysis data and observation data. The differences between reanalysis wind speed values and multi-point over-lake measurements are attributed to several reasons: 1) the coarse resolution of reanalysis data (Gualtieri, 2021); 2) surface roughness and topography differences between the assimilation scheme of the reanalysis datasets and the in-situ sites (Jourdiere, 2020). As described in our previous response, the wind function from McJannet et al. (2012) was developed for use with “land-based meteorological data”. Thus, gridded wind speed values from reanalysis datasets are appropriate for such calculation. It is also worth noting that the uncertainties from the reanalysis wind speeds have a relatively small impact on the total latent heat flux. This is because the energy-balance component is much larger than the aerodynamic component (Xiao et al., 2020). For instance, in the Priestley-Taylor equation, a coefficient of 1.26 is applied to the energy-balance component by assuming that the aerodynamic component is approximately equivalent to 26% of the energy-balance component.

Over Lake Mead, the energy-balance terms and the VPD from the reanalysis data all agree well with their counterpart observations. For wind speed, the reanalysis data are not as comparable to the over-lake observations due to 1) different surface roughness values (low lake roughness leads to higher wind speeds); and 2) different spatial resolutions (smoothed gridded data vs. more spatially heterogeneous point measurements). Together, the results shown in Fig. R1-2 and Fig. R1-3 suggest that the partitioning of the components in our algorithm is robust, and thus the good agreement of the latent heat flux simulation is not resulted from the error cancellation. Furthermore, these results confirm that the algorithm is applicable and reliable for large lakes (regardless if the lake is shallow or deep).

Fig. R1-2. Evaluation of a) net radiation, b) heat storage change, c) latent heat flux, d) vapor pressure deficit, and e) wind speed at Lake Taihu (average depth of 1.9m) at a monthly scale. The shaded area represents the estimation uncertainty from different input forcing datasets (i.e., TerraClimate, ERA5, and GLDAS) and the dashed line represents the average values. The wind speed from the reanalysis datasets have been converted from the 10m reanalysis height to the heights of the in-situ data measurements according to the wind profile power law.

Fig. R1-3. Evaluation of a) net radiation, b) heat storage change, c) latent heat flux, d) vapor pressure deficit, and e) wind speed at Lake Mead (average depth >50m) at a monthly scale. The shaded area represents the estimation uncertainty from different input forcing datasets (i.e., TerraClimate, ERA5, and GLDAS) and the dashed line represents the average values. The wind speed from the reanalysis datasets have been converted from the 10m reanalysis height to the heights of in-situ data measurements according to the wind profile power law.

We would also like to emphasize that the wind speed developed in McJannet et al. (2012) is an empirical function, which contains uncertainties from multiple sources, such as the “*curve-fitting process, measurement errors, upwind roughness, stability conditions, and extrapolation (when applicable)*” (see Zhao and Gao, 2019 for more detailed discussion on this). However, the validation results from this manuscript for lakes across different climatic and geographic regions

(Fig. S5, attached here as Fig. R1-4)—and for lakes with a large fetch (such as Lake Taihu in Fig. R1-1 and R1-2 and Lake Mead in Fig. R1-3)—suggested a satisfactory simulation performance for both the evaporation rate and energy balance terms. This also implies that the aerodynamic term is reliable. These satisfactory results are mainly attributed to the novel **heat storage quantification algorithm** that we developed in Zhao and Gao (2019). Without heat storage simulation, the evaporation rate from the original Penman equation has shown significant bias both in terms of magnitude and seasonality (Fig. R1-4). Given the evidence that we have listed above, we are confident in the global applicability of our evaporation rate algorithm, and in the results presented in this manuscript. These new results (Fig. R1-2 and R1-3) have been added to the Supplementary Information of our manuscript (Fig. S14 and S15) and the associated discussion has been added to the main text of the manuscript: “Furthermore, we have systematically evaluated the energy-balance and aerodynamic terms using observations at Lake Taihu (Lee et al., 2014; Zhang et al., 2020) (Supplementary Fig. 14) and Lake Mead (Moreo, 2015) (Supplementary Fig. 15). Our evaluations for both shallow and deep lakes indicate that our algorithm is robust, and that the good agreement of the evaporation rate simulation is not caused by the cancellation of errors.” (Lines 276-280).

Fig. R1-4. (Fig. S5) Evaporation rate validation for a) Lake Mead, USA, b) White Bear Lake, USA, c) Ross Barnett Reservoir, USA, d) Lake Calm, USA, e) Lake Five-O, USA, f) Lake Taihu, China, g) Lake Nasser, Egypt, h) Lake Kasumigaura, Japan, and i) Lake Kinneret, Israel.

R1C3. Confounding effect of evap rate and ice duration

With regard to the confounding effect of the increasing evaporation rate and decreasing lake ice, we have systematically evaluated it in our manuscript. The evaporation volume was

specifically calculated as the product of the evaporation rate, surface area, and fraction of ice duration (Eq. 1 in the main text). The increasing trend of evaporation volume (shown in Fig. 3d in the main text) is a result of the increasing evaporation rate (Fig. 3b), which is caused by warming, and the increasing open water area (Fig. 3c), which is caused by hydrological variation and decreasing lake ice (Xiao et al., 2018). In addition, we have evaluated how the variability of the evaporation volume is attributed to the changes of the three contributing factors: the evaporation rate, the surface area, and the fraction of lake ice duration (Fig. 4). Also, using the controlled experiments (see Lines 150-158), we have quantified that “The drivers of evaporation volume trend are the increasing evaporation rate, the decreasing lake ice coverage, and the increasing lake surface area, which contribute 58%, 23%, and 19%, respectively.” (Lines 31-32)

Reference

- 1) Gualtieri, G. (2021). Reliability of ERA5 Reanalysis Data for Wind Resource Assessment: A Comparison against Tall Towers. *Energies*, 14(14), 4169.
- 2) Jourdir, B. (2020). Evaluation of ERA5, MERRA-2, COSMO-REA6, NEWA and AROME to simulate wind power production over France. *Advances in Science and Research*, 17, 63-77.
- 3) Lee, X., Liu, S., Xiao, W., Wang, W., Gao, Z., Cao, C., ... & Zhang, M. (2014). The Taihu Eddy Flux Network: An observational program on energy, water, and greenhouse gas fluxes of a large freshwater lake. *Bulletin of the American Meteorological Society*, 95(10), 1583-1594.
- 4) McJannet, D. L., Webster, I. T., & Cook, F. J. (2012). An area-dependent wind function for estimating open water evaporation using land-based meteorological data. *Environmental Modelling & Software*, 31, 76-83.
- 5) Moreo MT. Evaporation Data from Lake Mead and Lake Mohave, Nevada and Arizona, March 2010 through April 2015: U.S. Geological Survey Data Release, <http://dx.doi.org/10.5066/F79C6VG3>. US Geological Survey (2015).
- 6) Wild, M., Ohmura, A., Schär, C., Müller, G., Folini, D., Schwarz, M., ... & Sanchez-Lorenzo, A. (2017). The Global Energy Balance Archive (GEBA) version 2017: A database for worldwide measured surface energy fluxes. *Earth System Science Data*, 9(2), 601-613.
- 7) Xiao, K., Griffis, T. J., Baker, J. M., Bolstad, P. V., Erickson, M. D., Lee, X., ... & Nieber, J. L. (2018). Evaporation from a temperate closed-basin lake and its impact on present, past, and future water level. *Journal of Hydrology*, 561, 59-75.
- 8) Xiao, W., Zhang, Z., Wang, W., Zhang, M., Liu, Q., Hu, Y., ... & Lee, X. (2020). Radiation controls the interannual variability of evaporation of a subtropical lake. *Journal of Geophysical Research: Atmospheres*, 125(8), e2019JD031264.
- 9) Zhang, Z., Zhang, M., Cao, C., Wang, W., Xiao, W., Xie, C., ... & Lee, X. (2020). A dataset of microclimate and radiation and energy fluxes from the Lake Taihu eddy flux network. *Earth System Science Data*, 12(4), 2635-2645.
- 10) Zhao, G., & Gao, H. (2019). Estimating reservoir evaporation losses for the United States: Fusing remote sensing and modeling approaches. *Remote Sensing of Environment*, 226, 109-124.

Reviewer #2 (Remarks to the Author):

I have three major concerns:

(1) Landsat-based global surface water database (GSWD) from Pekel et al. (2016) provides monthly frequency of water pixels mapped. If the pixels are cloud covered, the pixels will be replaced with cloud-free pixels in other time. Therefore, monthly pixels of waterbodies from GSWD are not real status of waterbody at a specific date. Pekel et al. (2016) showed this clear, and only use the waterbody frequency of more than 75% for trend examination. Especially, waterbody is not lake, many works need to do for lake extraction from waterbody. The authors used a global lake product from HydroLAKES, which is derived from SRTM DEM in February 2000 and includes 1.42 million lakes. The authors used the GSWD directly and claimed they examined evaporative water loss of 1.42 million global lakes. Obviously, the authors did not understand how the GSWD was produced, and lake mapping well, which can cause error estimate of the trend of the evaporative volume for global lakes. I do not think the authors can address this problem appropriately.

(2) The error data of lakes (concept) used can result in error conclusion such as increasing lake surface area accounts for 19% of global lake evaporative water loss.

(3) Lake ice phenology was estimated using air temperature and freeze/thaw lags. I do not think this is a reliable estimate as which has no physical explanation and air temperature has large uncertainties in station sparse regions.

Based these major considerations, I do not think the authors provide a reliable estimate of global lake evaporative water loss, and attribution analysis such as decreasing lake ice coverage and the increasing lake surface area.

We would like to thank the reviewer again for your time in reviewing our revised manuscript. However, we do not agree with the statements about the usage of GSWD, HydroLAKES, and the simulation of lake ice phenology. While we have explained these in detail in the last set of responses, we would like to add more explanation here. In addition, we have listed multiple papers to demonstrate that we are fully knowledgeable about how these datasets were produced, and how to properly use them to generate a consistent and high-quality evaporation dataset for global lakes.

R2C1. Usage of GSWD and HydroLAKES

The reviewer may have misunderstood our novel approach for generating the monthly area time series **for each lake**. Our algorithm is built upon our previously published and well-validated paper (Zhao and Gao, 2018), which specifically addresses the contamination and misclassification issues of Landsat images and the following GSWD (Pekel et al., 2016). Also, it

is true that GSWD is a “waterbody” product (Zhang et al., 2020). This is why we used HydroLAKES (Messenger et al., 2016) as the “lake” masks to extract the area dynamics from GSWD as an additional step (to ensure that the extracted water pixels are from lakes). Indeed, this procedure is similar to that using the lake inventory for the lake outlines in Zhang et al. (2019), except that we used HydroLAKES for lake masks. We have added new clarification and new references (of these two studies) to the revised manuscript: “The GSWD is a global “waterbody” product (Zhang et al., 2020) that was derived from Landsat imagery via an expert system that considers numerous factors (e.g., cloud shadow, terrain shadow, lava). However, remotely sensed “waterbody” does not represent the “lake” water extent. Thus, we further used HydroLAKES shapefiles as the outer boundaries of lakes to ensure that the extracted water pixels from GSWD are from lakes. This type of approach has been commonly adopted for lake area time series estimations (Klein et al., 2021; Meyer et al., 2020; Zhang et al., 2019).” (Lines 292-298). In addition, we did not just use the “monthly frequency” from GSWD to create the monthly time series. Instead, we integrated the “yearly” product and the “monthly frequency” from GSWD. The “yearly” product provides the inter-annual variability of lake surface area. For EACH YEAR, the “monthly frequency” product provides the monthly variability. We have explicitly stated such an approach in our manuscript (“Lake surface area dynamics” subheading in the “Methods” section).

- 1) Zhao, G., & Gao, H. (2018). Automatic correction of contaminated images for assessment of reservoir surface area dynamics. *Geophysical Research Letters*, 45(12), 6092-6099.
- 2) Pekel, J. F., Cottam, A., Gorelick, N., & Belward, A. S. (2016). High-resolution mapping of global surface water and its long-term changes. *Nature*, 540(7633), 418-422.
- 3) Messenger, M. L., Lehner, B., Grill, G., Nedeva, I., & Schmitt, O. (2016). Estimating the volume and age of water stored in global lakes using a geo-statistical approach. *Nature communications*, 7(1), 1-11.
- 4) Zhang, G., Yao, T., Chen, W., Zheng, G., Shum, C. K., Yang, K., ... & Jia, Y. (2019). Regional differences of lake evolution across China during 1960s–2015 and its natural and anthropogenic causes. *Remote Sensing of Environment*, 221, 386-404.
- 5) Zhang, G., Yao, T., Xie, H., Yang, K., Zhu, L., Shum, C. K., ... & Ke, C. (2020). Response of Tibetan Plateau lakes to climate change: Trends, patterns, and mechanisms. *Earth-Science Reviews*, 208, 103269.
- 6) Klein, I., Mayr, S., Gessner, U., Hirner, A., & Kuenzer, C. (2021). Water and hydropower reservoirs: High temporal resolution time series derived from MODIS data to characterize seasonality and variability. *Remote Sensing of Environment*, 253, 112207.
- 7) Meyer, M. F., Labou, S. G., Cramer, A. N., Brousil, M. R., & Luff, B. T. (2020). The global lake area, climate, and population dataset. *Scientific data*, 7(1), 1-12.

1.1. Similar usage of GSWD from Pekel et al., (2016)

The time series data (both monthly and annual) from GSWD have been widely used for the evaluation of lake area dynamics. It is worth noting that the developer of the GSWD dataset (Dr. Pekel) is a coauthor for several of these papers. For example, Busker et al., (2019) generated the monthly reservoir storage using the monthly area product from GSWD. Bastin et al. (2019)

used the yearly area product from GSWD to evaluate the long-term trends of permanent and seasonal water within protected areas globally. Similarly, Mentaschi et al. (2018) used the yearly area product from GSWD to quantify the long-term change of global coastlines. A study conducted by Meyer et al. (2020) is similar to our area quantification work, although they only provided data at an annual time scale. The authors combined the boundaries from **HydroLAKES** with the **GSWD** yearly product to generate the annual lake area dynamics for 1.42 million lakes.

- 1) Busker, T., de Roo, A., Gelati, E., Schwatke, C., Adamovic, M., Bisselink, B., **Pekel, J.F.** and Cottam, A. (2019). A global lake and reservoir volume analysis using a surface water dataset and satellite altimetry. *Hydrology and Earth System Sciences*, 23(2), 669-690.
- 2) Bastin, L., Gorelick, N., Saura, S., Bertzky, B., Dubois, G., Fortin, M. J., & **Pekel, J. F.** (2019). Inland surface waters in protected areas globally: Current coverage and 30-year trends. *PloS One*, 14(1), e0210496.
- 3) Mentaschi, L., Vousdoukas, M. I., **Pekel, J. F.**, Voukouvalas, E., & Feyen, L. (2018). Global long-term observations of coastal erosion and accretion. *Scientific Reports*, 8(1), 1-11.
- 4) **Meyer, M. F.**, Labou, S. G., Cramer, A. N., Brousil, M. R., & Luff, B. T. (2020). The global lake area, climate, and population dataset. *Scientific Data*, 7(1), 1-12.

1.2. Usage of Zhao and Gao (2018) [dataset and/or algorithm]

The lake surface area data in our manuscript is based on the algorithm developed in Zhao and Gao (2018) with improvements in computational efficiency and accuracy for data-limited years (see R2C1 for more detail in our previous response). Our algorithm and the associated long-term monthly reservoir area product in Zhao and Gao (2018) has been widely used in the hydrological and remote sensing communities. Perhaps the most noticeable case is in Keller et al. (2021), who primarily relied on our monthly area product to quantify the carbon emissions from reservoirs. Similarly, Tian et al. (2022) also directly used our area product to quantify the reservoir evaporation volume. In Biswas et al. (2021), our product was used to validate a newly developed area monitoring framework at a monthly time step. For the algorithm usage, Fang et al. (2019) and Feng et al. (2022) directly used our area enhancement algorithm to generate their monthly lake surface area time series.

- 1) Zhao, G., & Gao, H. (2018). Automatic correction of contaminated images for assessment of reservoir surface area dynamics. *Geophysical Research Letters*, 45(12), 6092-6099.
- 2) Keller, P. S., Marcé, R., Obrador, B., & Koschorreck, M. (2021). Global carbon budget of reservoirs is overturned by the quantification of drawdown areas. *Nature Geoscience*, 14(6), 402-408.
- 3) Tian, W., Liu, X., Wang, K., Bai, P., Liu, C., & Liang, X. (2022). Estimation of Global Reservoir Evaporation Losses. *Journal of Hydrology*, 127524.
- 4) Biswas, N. K., Hossain, F., Bonnema, M., Lee, H., & Chishtie, F. (2021). Towards a global Reservoir Assessment Tool for predicting hydrologic impacts and operating patterns of existing and planned reservoirs. *Environmental Modelling & Software*, 140, 105043.

- 5) Fang, Y., Li, H., Wan, W., Zhu, S., Wang, Z., Hong, Y., & Wang, H. (2019). Assessment of water storage change in China's lakes and reservoirs over the last three decades. *Remote Sensing*, 11(12), 1467.
- 6) Feng, Y., Zhang, H., Tao, S., Ao, Z., Song, C., Chave, J., ... & Fang, J. (2022). Decadal Lake Volume Changes (2003–2020) and Driving Forces at a Global Scale. *Remote Sensing*, 14(4), 1032.

1.3. Usage of HydroLAKES to extract water dynamics

We believe that the reviewer has misunderstood our approach of using HydroLAKES to extract lake areas from GSWD. Indeed, such approach has been well adopted in the field of remote sensing of lakes, both in our own publications and those from other colleagues (see references below). We would like to emphasize that HydroLAKES is a global dataset that contains 1.42 million lakes, and we used it to provide the lake masks to extract area time series from GSWD. In **Meyer et al. (2020)**, the authors used a similar approach that combined boundaries from **HydroLAKES** with the **GSWD** yearly product to generate the annual lake area dynamics for 1.42 million lakes. Other similar examples include Klein et al. (2021), Khandelwal et al. (2017), Tortini et al. (2020), Ling et al. (2020), Yao et al. (2019), and Deng et al. (2020).

- 1) **Zhao, G., & Gao, H.** (2018). Automatic correction of contaminated images for assessment of reservoir surface area dynamics. *Geophysical Research Letters*, 45(12), 6092-6099.
- 2) **Li, Y., Gao, H., Zhao, G., & Tseng, K. H.** (2020). A high-resolution bathymetry dataset for global reservoirs using multi-source satellite imagery and altimetry. *Remote Sensing of Environment*, 244, 111831.
- 3) **Li, Y., Zhao, G., Shah, D., Zhao, M., Sarkar, S., Devadiga, S., ... & Gao, H.** (2021). NASA's MODIS/VIIRS Global Water Reservoir Product Suite from Moderate Resolution Remote Sensing Data. *Remote Sensing*, 13(4), 565.
- 4) Meyer, M. F., Labou, S. G., Cramer, A. N., Brousil, M. R., & Luff, B. T. (2020). The global lake area, climate, and population dataset. *Scientific data*, 7(1), 1-12. (HydroLAKES is directly used as the mask file)
- 5) Klein, I., Mayr, S., Gessner, U., Hirner, A., & Kuenzer, C. (2021). Water and hydropower reservoirs: High temporal resolution time series derived from MODIS data to characterize seasonality and variability. *Remote Sensing of Environment*, 253, 112207. (HydroLAKES is directly used as the mask file)
- 6) Tortini, R., Noujdina, N., Yeo, S., Ricko, M., Birkett, C. M., Khandelwal, A., ... & Lettenmaier, D. P. (2020). Satellite-based remote sensing data set of global surface water storage change from 1992 to 2018. *Earth System Science Data*, 12(2), 1141-1151. (GRanD, which is a subset of HydroLAKES is used as the mask file)
- 7) Khandelwal, A., Karpatne, A., Marlier, M. E., Kim, J., Lettenmaier, D. P., & Kumar, V. (2017). An approach for global monitoring of surface water extent variations in reservoirs using MODIS data. *Remote sensing of Environment*, 202, 113-128. (GRanD, which is a subset of HydroLAKES is used as the mask file)
- 8) Ling, F., Li, X., Foody, G. M., Boyd, D., Ge, Y., Li, X., & Du, Y. (2020). Monitoring surface water area variations of reservoirs using daily MODIS images by exploring sub-pixel information. *ISPRS Journal of Photogrammetry and Remote Sensing*, 168, 141-152. (GRanD, which is a subset of HydroLAKES is used as the mask file)
- 9) Yao, F., Wang, J., Wang, C., & Crétaux, J. F. (2019). Constructing long-term high-frequency time series of global lake and reservoir areas using Landsat imagery. *Remote Sensing of Environment*, 232, 111210. (circa-2000, which is similar to HydroLAKES, is used as the mask file)

10) Deng, X., Song, C., Liu, K., Ke, L., Zhang, W., Ma, R., ... & Wu, Q. (2020). Remote sensing estimation of catchment-scale reservoir water impoundment in the upper Yellow River and implications for river discharge alteration. *Journal of Hydrology*, 585, 124791. (JRC max_extent, which is similar to HydroLAKES, is used as the mask file)

R2C2. Accuracy of lake area estimations

As evidenced in R2C1, we have properly handled the lake area estimations using GSWD, using a well-recognized concept in the field. In our previous response, we provided comprehensive validation results (attached here as Fig. R2-1 and Fig. R2-2) for this method. The results from this study are shown as solid black lines, which match well with observed storage or elevation dynamics. Please refer to our previous response in R2C1 for more detail.

Fig. R2-1. Validation of reconstructed monthly area (this study) using observed reservoir storage or elevation time series. The time series from GRSAD is also plotted for reference. The two values in the parenthesis represent the R^2 values for 1) GRSAD area and observed storage or elevation and 2) reconstructed area and observed storage or elevation.

Fig. R2-2. Validation of reconstructed monthly area values (this study) using satellite altimetry based elevation time series data. The value in the parenthesis represents the R^2 values for the reconstructed area and the altimetry elevation.

R2C3. Usage of air temperature to simulate river/lake ice

We would like to emphasize that air temperature is a widely used proxy, and the most important driver, for river and lake ice duration simulations (Yang et al., 2020 Nature; Sharma et al., 2019 Nature Climate Change; Woolway et al., 2020 Nature Reviews Earth & Environment). An air temperature-based method is markedly valid for lakes that have small surface areas (Smits et al., 2021; Caldwell et al., 2021). For large lakes, the heat storage in the summer, and average winter temperature, can notably impact the freeze-up and break-up timing. Thus, we modeled 1) the freeze-lag using average lake depth and 2) thaw-lag using average winter temperature (Extended Data Fig. 3). We have provided systematic validation of this method in the response for the previous round (see R2C2 in our previous response).

Here we would like to list the evidence that air temperature has been used as the major driver for river/lake ice simulation. For example, Yang et al. (2020) used air temperature alone to calculate the probability of river ice. In Sharma et al. (2019), the authors used air temperature as the primary driver and combined it with lake characteristics to train a classification tree to calculate lake ice duration. More specifically, Arp et al. (2013) emphasized that using the simple 0°C air temperature isotherm method together with lake area can explain 80% of the ice-out timing variation.

- 1) Yang, X., Pavelsky, T. M., & Allen, G. H. (2020). The past and future of global river ice. *Nature*, 577(7788), 69-73.
- 2) Sharma, S., Blagrove, K., Magnuson, J. J., O'Reilly, C. M., Oliver, S., Batt, R. D., ... & Woolway, R. I. (2019). Widespread loss of lake ice around the Northern Hemisphere in a warming world. *Nature Climate Change*, 9(3), 227-231.
- 3) Arp, C. D., Jones, B. M., & Grosse, G. (2013). Recent lake ice-out phenology within and among lake districts of Alaska, USA. *Limnology and Oceanography*, 58(6), 2013-2028.
- 4) Woolway, R. I., Kraemer, B. M., Lenters, J. D., Merchant, C. J., O'Reilly, C. M., & Sharma, S. (2020). Global lake responses to climate change. *Nature Reviews Earth & Environment*, 1(8), 388-403.
- 5) Smits, A. P., Gomez, N. W., Dozier, J., & Sadro, S. (2021). Winter Climate and Lake Morphology Control Ice Phenology and Under-Ice Temperature and Oxygen Regimes in Mountain Lakes. *Journal of Geophysical Research: Biogeosciences*, 126(8), e2021JG006277.
- 6) Caldwell, T. J., Chandra, S., Albright, T. P., Harpold, A. A., Dilts, T. E., Greenberg, J. A., ... & Dettinger, M. D. (2021). Drivers and projections of ice phenology in mountain lakes in the western United States. *Limnology and Oceanography*, 66(3), 995-1008.

Reviewer #3 (Remarks to the Author):

The authors have thoroughly addressed my comments, I and believe those of the other reviewers. I have no further edits or comments. Only that the authors should consider making their code publicly available (e.g. github etc.) if possible. I recommend this for publication.

Thank you very much, Dr. Gardner! We will publish the code via GitHub once the manuscript is accepted.

Reviewers' Comments:

Reviewer #1:

Remarks to the Author:

I am happy with the performance evaluation carried out by the authors against two new lake datasets. I have no additional comments and support publication of this paper in NC.

Reviewer #2:

Remarks to the Author:

The authors have improved this manuscript greatly. Before the possible publication this manuscript, some obvious issues still need to be improved. After these revisions, I agree with the acceptance of this manuscript.

1. Title: I suggest change "lakes" to "lakes and reservoirs" or "water bodies" as the reservoirs are different from lakes, can not only defined as artificial lakes. Moreover, in the manuscript, the reservoirs are used many times.
2. The writing of this manuscript still need to improve. The methods and results are mixed appearance in Results section. The Results section should only present results directly and clearly.
3. The uncertainties (plus/minus) should be provided in all estimates (trend and %).
4. L62: lake evaporation volume should be defined at the first appearance.
5. L76: it is average from 1985 to 2018? While you mention the change, the time period is necessary.
5. The land *ET* volume was quantified by using the MODIS product. How this is quantified before 2000 while MODIS is no available.
6. L271: ...average lake depth after Patalas. How about this lake depth data compared with another lake depth data from Messenger et al., 2016 (reference 31)? Why you selected Patalas data here?
7. L288: What is "not shown" here? Why it is not shown?
8. The authors mentioned the lake ice data cannot be validated by MODIS lake ice products as cloud cover. This is not a reasonable excuse. The Landsat data for deriving water bodies extents are also optical images and are susceptible to cloud contamination. The authors can use the Landsat images to validate/compare the lake ice phenology. At least, it can provide some evidences that if the estimate used by the authors is matched.
9. The evaporation rate estimation method cannot be applied to ice-covered season for ice-sublimation estimation. And ice-covered seasons covering a long-period each year in high-altitude and high-latitude lakes and the sublimation values should not be ignored. The uncertainties arising from ice sublimation should be considered more physically. More discussions for these could be included.

Reviewer #1 (Remarks to the Author):

I am happy with the performance evaluation carried out by the authors against two new lake datasets. I have no additional comments and support publication of this paper in NC.

R1C0: We greatly appreciate your time and effort in reviewing our manuscript, which have truly improved the quality our work.

Reviewer #2 (Remarks to the Author):

The authors have improved this manuscript greatly. Before the possible publication this manuscript, some obvious issues still need to be improved. After these revisions, I agree with the acceptance of this manuscript.

R2C0: We are grateful to you again for reviewing our manuscript. We have edited the manuscript accordingly based on the reviewer's comments.

1. Title: I suggest change "lakes" to "lakes and reservoirs" or "water bodies" as the reservoirs are different from lakes, can not only defined as artificial lakes. Moreover, in the manuscript, the reservoirs are used many times.

R2C1: Thank you for the comment. After putting a lot of thought into the revision of the title and fully discussing with coauthors, we chose to keep the word "lakes". The decision is based on the following considerations.

- 1) We would like to be consistent with the terms used by the original HydroLAKES dataset (Messenger et al., 2016), from which our work is built upon. In Messenger et al. (2016), they defined "lakes" as the total of natural lakes and human-made reservoirs.
- 2) Thus, if we change the title to "lakes and reservoirs", there would be underlying duplication. Also, because we used "lakes" to refer all natural and artificial lakes in the manuscript, such change will also introduce inconsistency between the title and the content.
- 3) We have used "water bodies" in our manuscript when referring to predefined lakes, yet "water bodies" is a more general term that can represent rivers, lakes, estuaries, bay, or oceans (https://en.wikipedia.org/wiki/Body_of_water).

We totally agree with you that clear definition is essential for our paper. Thus, to clearly distinguish the different terms that are used in our manuscript ("lakes", "natural lakes", "artificial lakes", and "reservoirs"), we have added explicit definitions in the abstract and introduction section (Lines 21, 32, 57). We believe such clarification can better deliver the message of our findings.

Reference

Messenger, M. L., Lehner, B., Grill, G., Nedeva, I., & Schmitt, O. (2016). Estimating the volume and age of water stored in global lakes using a geo-statistical approach. *Nature Communications*, 7(1), 1-11.

2. The writing of this manuscript still need to improve. The methods and results are mixed appearance in Results section. The Results section should only present results directly and clearly.

R2C2: The Results section has been revised by moving the sentences that are related to methods into the Methods section (Lines 156-159, Lines 428-435).

3. The uncertainties (plus/minus) should be provided in all estimates (trend and %).

R2C3: We have added uncertainties (plus/minus) for all major values (e.g., global average evaporation and global trend of evaporation), along with details about the uncertainty analysis methods (Lines 449-463 and Supplementary Fig. 3). The values associated with regional analyses (e.g., Lines 131, 133) were provided as single values to improve the readability of the manuscript.

4. L62: lake evaporation volume should be defined at the first appearance.

R2C4: Thank you. GLEV is used as an acronym for the “global lake evaporation volume” dataset, and the definition of lake evaporation volume is explained as “evaporative water loss” in the second half of this sentence.

5. L76: it is average from 1985 to 2018? While you mention the change, the time period is necessary.

R2C5: Thank you. We have added the temporal range (i.e., 1985 to 2018) to the sentence.

5. The land *ET* volume was quantified by using the MODIS product. How this is quantified before 2000 while MODIS is no available.

R2C5.1: Because the purpose for the analysis in Fig. 2 is to quantify the long-term average percentage of lake evaporation over total ET, we used the overlapping years between MODIS and our product (i.e., 2001 to 2018). This information can be found in the caption of Fig. 2. The year of 2000 was excluded because MODIS Terra started data delivery in Feb 2000.

6. L271: ...average lake depth after Patalas. How about this lake depth data compared with another lake depth data from Messenger et al., 2016 (reference 31)? Why you selected Patalas data here?

R2C6: We have clarified that Patalas provides the method employed to calculate the potential epilimnion thickness using surface area data (Patalas, 1984; Bohrer and Schultze, 2008). The actual epilimnion thickness was determined as the smaller value between the potential epilimnion thickness and the averaged lake depth from the HydroLAKES dataset (Messenger et al., 2016) (Lines 278-280).

Reference

Bohrer, B., & Schultze, M. (2008). Stratification of lakes. *Reviews of Geophysics*, 46(2).

Patalas, K. (1984). Mid-summer mixing depths of lakes of different latitudes. *Internationale Vereinigung für theoretische und angewandte Limnologie: Verhandlungen*, 22(1), 97-102.

Messenger, M. L., Lehner, B., Grill, G., Nedeva, I., & Schmitt, O. (2016). Estimating the volume and age of water stored in global lakes using a geo-statistical approach. *Nature Communications*, 7(1), 1-11.

7. L288: What is “not shown” here? Why it is not shown?

R2C7: We have removed the “not shown” to avoid confusion. The reason why the comparison was not shown in our manuscript is because it is a comparison between our results—Fig. S1a—with a figure from another study (i.e., Fig. 2b from Wang et al., 2018), to which we do not own the copyright. More detailed discussion, with a side-by-side comparison of these two figures, can be found in R1C12 from the first round.

Reference

Wang, W., Lee, X., Xiao, W., Liu, S., Schultz, N., Wang, Y., ... & Zhao, L. (2018). Global lake evaporation accelerated by changes in surface energy allocation in a warmer climate. *Nature Geoscience*, 11(6), 410-414.

8. The authors mentioned the lake ice data cannot be validated by MODIS lake ice products as cloud cover. This is not a reasonable excuse. The Landsat data for deriving water bodies extents are also optical images and are susceptible to cloud contamination. The authors can use the Landsat images to validate/compare the lake ice phenology. At least, it can provide some evidences that if the estimate used by the authors is matched.

R2C8: Thank you for the comment. We agree that both the MODIS and Landsat products have great potential to derive lake ice phenology. However, the derivation of a reliable lake ice phenology time series from either MODIS or Landsat would need novel algorithm development to address the limitation of both sensors: 1. MODIS has a coarse spatial resolution (250m to 1000m depending on the band; NDSI has a resolution of 500m) and thus cannot accurately resolve the ice coverage for relatively small lakes (e.g., lakes $\leq 25 \text{ km}^2$; 100 pixels of NDSI). 2. Landsat has a coarse temporal resolution (16-day) that limits the detection of ice-on and ice-off dates. Additionally, both sensors are susceptible to cloud cover and atmospheric conditions, leading to omission and commission errors for lake ice detection (Yang et al., 2021; Wu et al., 2021). Thus, in this study, we used in-situ observed ice-on and ice-off data to validate the accuracy of our calculation (Supplementary Fig. 10).

While the derivation of the ice fraction time series values from MODIS or Landsat is beyond the scope of our current study, we provided an example to show how our calculated fraction of ice duration compares with each (Fig. R2C8, added in the manuscript as Supplementary Fig. 14). The Landsat ice fraction time series were derived following three steps:

- 1) For each Landsat image that intersects with the given lake, “clear water area” was extracted by applying the water mask from GSWD onto the cloud-and-shadow-removed Landsat image.

- 2) The “clear ice-on-water area” was then extracted by overlapping the “clear water area” with the snow/ice band of the Landsat image that was derived based on the CFMask algorithm.
- 3) If the “clear water area” was larger than 50% of the GSWD water mask, we calculated the lake ice fraction using the “clear ice-on-water area” divided by the “clear water area”. This step is used to assure the quality of the lake ice fraction calculation.

Due to the coarse temporal resolution and cloud contamination issues for Landsat images, Landsat-based ice fraction only provides scattered values (Fig. R2C8). However, for the periods that have sufficient data (e.g., year 2005), it shows the good agreement with our simulation (bias is +2 days) and with in-situ observations (-5 days). For the years that have limited data (e.g., year 2006), the bias of the Landsat-based ice fraction can be large (-40 days). We have added new results pertaining to this in the Supplementary Information as Supplementary Fig. 14. In addition, we have added discussion to highlight the great potential of using satellite-based lake ice phenology data in the future in the main text (Lines 423-425).

Fig. R2C8. Comparison of in-situ observed (Benson et al., 2012), MODIS (MOD10A2), Landsat (CFMask), and simulated lake ice phenology for Lake Oneida, USA.

Reference

- Yang, X., Pavelsky, T. M., Bendezu, L. P., & Zhang, S. (2021). Simple Method to Extract Lake Ice Condition From Landsat Images. *IEEE Transactions on Geoscience and Remote Sensing*.
- Wu, Y., Duguay, C. R., & Xu, L. (2021). Assessment of machine learning classifiers for global lake ice cover mapping from MODIS TOA reflectance data. *Remote Sensing of Environment*, 253, 112206.

9. The evaporation rate estimation method cannot be applied to ice-covered season for ice-sublimation estimation. And ice-covered seasons covering a long-period each year in high-altitude and high-latitude lakes and the sublimation values should not be ignored. The uncertainties arising from ice sublimation should be considered more physically. More discussions for these could be included.

R2C9: Thank you. We have added the discussion about lake ice sublimation in the “Sources of uncertainty and algorithm caveats” section (Lines 469-472). In addition, we have also added the full discussion—including the snow-ice interactions, and measurements of the ice sublimation rate—into the Supplementary Information (Supplementary Note 1). We highlighted the fact that the ice sublimation can also be a source of lake water loss (although it is likely to be much smaller than water evaporation on a global scale).